# Fiber-Type Shifting in Sarcopenia of Old Age: Proteomic Profiling of the Contractile Apparatus of Skeletal Muscles

**DOI:** 10.3390/ijms24032415

**Published:** 2023-01-26

**Authors:** Paul Dowling, Stephen Gargan, Dieter Swandulla, Kay Ohlendieck

**Affiliations:** 1Department of Biology, Maynooth University, National University of Ireland, W23 F2H6 Maynooth, Co. Kildare, Ireland; 2Kathleen Lonsdale Institute for Human Health Research, Maynooth University, W23 F2H6 Maynooth, Co. Kildare, Ireland; 3Institute of Physiology, University of Bonn, D53115 Bonn, Germany

**Keywords:** actin, aging, atrophy, frailty, myosin, sarcomere, sarcopenia, tropomyosin, troponin

## Abstract

The progressive loss of skeletal muscle mass and concomitant reduction in contractile strength plays a central role in frailty syndrome. Age-related neuronal impairments are closely associated with sarcopenia in the elderly, which is characterized by severe muscular atrophy that can considerably lessen the overall quality of life at old age. Mass-spectrometry-based proteomic surveys of senescent human skeletal muscles, as well as animal models of sarcopenia, have decisively improved our understanding of the molecular and cellular consequences of muscular atrophy and associated fiber-type shifting during aging. This review outlines the mass spectrometric identification of proteome-wide changes in atrophying skeletal muscles, with a focus on contractile proteins as potential markers of changes in fiber-type distribution patterns. The observed trend of fast-to-slow transitions in individual human skeletal muscles during the aging process is most likely linked to a preferential susceptibility of fast-twitching muscle fibers to muscular atrophy. Studies with senescent animal models, including mostly aged rodent skeletal muscles, have confirmed fiber-type shifting. The proteomic analysis of fast versus slow isoforms of key contractile proteins, such as myosin heavy chains, myosin light chains, actins, troponins and tropomyosins, suggests them as suitable bioanalytical tools of fiber-type transitions during aging.

## 1. Introduction

The loss of skeletal muscle mass and contractile strength can be induced by the lack of suitable physical activity levels, extended periods of disuse or disease [1,2,3]. Acute forms of skeletal muscle wasting are often observed during physical trauma and sepsis [4]. Many chronic conditions are also associated with muscular atrophy, including cancer cachexia, congestive heart failure, diabetes mellitus, chronic obstructive pulmonary disease, glucocorticoid-induced Cushing syndrome, malnutrition, long-lasting infections, acquired immunodeficiency syndrome and kidney failure [5,6,7]. Chronic diseases triggering motor neuron abnormalities, such as amyotrophic lateral sclerosis, are a major clinical cause of muscular atrophy [8]. However, the most common form of contractile fiber wasting in association with muscular atrophy is represented by systemic changes during sarcopenia of old age [9,10,11].

Atrophying skeletal muscles are a major feature of the aging phenotype in humans [12], and often the degree of contractile weakness is even more pronounced than the extent of lost muscle mass [13,14,15]. Sarcopenia of old age is closely connected to frailty [16], as well as an increased frequency of falls and fractures [17,18,19], resulting in a drastically reduced quality of life in the elderly [20] that are affected by substantial skeletal muscle wasting [21]. Reduced skeletal muscle tissue mass in conjunction with low gait speed are typical indicators of sarcopenia [22], whereby the clinical definition of sarcopenia [23] relates to a significantly reduced percentage of muscle tissue quantity and/or quality as compared to the mean determined in younger and healthy adults of similar ethnic background and the same gender [24]. Variations in contractile strength due to aging can be conveniently determined by a variety of performance tests that evaluate physical parameters such as walking ability, gait speed, grip strength, standing capability and stair climbing [25,26,27]. However, the histo-morphometric characterization of the aging human musculature indicates significant differences in the degree of the structural decline in individual skeletal muscles [28].

Thus, for a deeper mechanistic understanding of the aging process and frailty syndrome, it is crucial to study aging-related changes at the level of systems biology [29,30], including the role of cellular stress, mitochondrial abnormalities, disturbed ion handling, impaired protein metabolism and epigenetic changes that may adversely affect tissue integrity and thus cause disturbed bioenergetic pathways, abnormal proteostasis, hormonal imbalances, impaired ion homeostasis and reduced neuromuscular activity [31,32,33]. The degree to which reduced fitness and higher risk for disease acquisition in association with somatic damage accumulation trigger cellular aging is intensely debated, versus the effects of adaptive processes on the development of senescence [34,35,36]. Since a variety of factors play a crucial role in promoting frailty, it is important to better understand the interplay between the dysregulation of central biochemical pathways, cellular signaling cascades and physiological systems [37]. This might lead to a more comprehensive idea of how a combination of chronic inflammation, metabolic syndrome, visceral obesity, insulin resistance, neurodegeneration and progressive skeletal muscle wasting negatively affects the general health status of the elderly [38].

Of central importance for muscle biogerontology is the determination of proteome-wide alterations in the aged organism [39,40] and the application of this biochemical knowledge to improve the treatment of frailty and muscular atrophy [41,42,43]. Proteomics is a key technology of modern biosciences [44] and crucial for advances in pharmacological research and biotechnology [45], as well as biomarker discovery [46]. Mass spectrometry is an ideal bioanalytical method for studying the molecular and cellular mechanisms that underlie normal physiological and biochemical processes, adaptive responses to changed functional demands and dysregulated mechanisms in the diseased state [47]. This includes biomolecular investigations into the multi-factorial triggering mechanisms involved in the general aging process of humans [48], and particularly frailty syndrome in the elderly [49]. 

This review summarizes the findings of major proteomic surveys of aged human skeletal muscles and relates them to the analysis of animal models of sarcopenia of old age. The main focus is on the mass spectrometric identification of contractile proteins as potential markers of muscle-fiber-type shifting [50,51,52]. Following an overview of proteomics as a highly useful bioanalytical tool to study skeletal muscle biology, this article discusses how biochemical and proteomic knowledge might be helpful to better understand the complexity of the neuromuscular aging process. The outline of the methodological approaches includes a description of the importance of two-dimensional gel electrophoresis for top-down proteomics, various antibody-based techniques, sample preparation for proteomic analysis, protein digestion for peptide mass spectrometry, key mass spectrometric methods, data acquisition for mass spectrometry and recent developments in single-cell proteomics and aptamer-based proteomics. The described biochemical surveys of fast versus slow isoforms of myosin, actin, troponin and tropomyosin suggest that isoform switching of these abundant muscle proteins is a suitable and robust process that can be utilized for muscle-fiber typing during age-related muscular atrophy. The final section of this review briefly summarizes the main factors that are involved in age-related muscular atrophy and associated fast-to-slow fiber-type shifting, and describes recent progress in biomarker discovery for monitoring muscle aging and the development of novel therapeutic approaches to treat sarcopenia of old age.

## 2. Proteomic Profiling of Skeletal Muscle Tissues

### 2.1. Proteomic Analysis Platforms and Associated Biochemical and Cell Biological Methodology

Following the establishment of the concept of the proteome [53] and mass-spectrometry-based proteomics as a highly useful screening tool in the modern biosciences [54], there has been a steady improvement of sample preparation, mass spectrometric instrumentation and data analysis pipelines using both bottom-up [55,56,57] and top-down proteomic techniques [58,59,60]. Importantly, modern biochemical analyses focus on the unifying concept of dynamic proteoforms being the basic units of protein activity [61,62,63]. This has given unprecedented insights into protein diversity and the role of proteins in cellular functions [64], including skeletal muscle tissues [65,66,67,68]. Advances in the field of proteomics now allow researchers to comprehensively study proteins expressed by an organism or biological system associated with physiological and pathophysiological phenotypes [44]. High-throughput technologies and more precision-based methodologies are now available to identify proteins and their modifications in complex samples [69,70,71,72]. This wide-ranging approach provides a solid platform to understand protein function in a particular biological pathway, and when perturbed, how this affects the biological system [73]. Consequently, proteomics has major applications in medicine and drug development [45,46,47]. The international HUPO Project has made enormous progress in establishing and cataloguing the highly dynamic human proteome [74,75,76], which forms the scientific basis of understanding protein homeostasis at the level of systems biology [77,78,79]. 

#### 2.1.1. Two-Dimensional Gel Electrophoresis

Two-dimensional gel electrophoresis (2D-GE) is a classic and commonly used method for proteome analysis [80,81,82] and presents an ideal bioanalytical approach for optimum protein separation prior to the systematic mass spectrometric profiling of proteoforms [61]. Although current proteomic analyses use mostly gel-free systems for the initial protein separation step, 2D-GE has not been superseded by chromatographical techniques for specialized applications in top-down proteomics [80,82]. 2D-GE is still a highly useful protein separation method that plays a key role in many proteomics analysis pipelines that focus on the identification and characterization of isolated and intact proteoforms [52,61]. The 2D-GE-based separation step is especially beneficial in the field of applied myology for analyzing the highly diverse array of isoforms of contractile proteins [65,66,67]. The large-scale survey of skeletal muscle proteins can be carried out under both native or denaturing conditions [66], including the thorough separation of key contractile proteins [52]. In the most frequently employed version of the 2D-GE technique, mixtures of proteins are separated by charge (based on the isoelectric point, p*I*, of individual proteins) in the first dimension, and by sodium dodecyl sulfate polyacrylamide slab gel electrophoresis (SDS-PAGE), which discriminates proteins based on their molecular weight, in the second dimension [83,84,85]. This approach can be used to separate several thousand different proteins on one 2D-gel [86,87,88]. Of note, the recently described micro-needling of the first-dimension gel can be used to considerably shorten the time requirements for the initial isoelectric focusing step in 2D-GE [89]. 

Most 2D-GE approaches are based on the usage of high concentrations of sodium dodecyl sulfate (SDS) for optimum solubilization of proteins in the second dimension [90,91], but 2D-GE can also be carried out with combinations of alternative detergents to increase the resolution of integral membrane proteins [92]. For example, a combination of the cationic detergent named benzyldimethyl-n-hexadecylammonium chloride (BAC) in the first dimension and SDS detergent in the second dimension is used for the BAC/SDS-PAGE technique [93]. Two-dimensional blue native polyacrylamide gel electrophoresis, usually referred to as BN-PAGE [94], separates proteins under native conditions [95] and is frequently used to characterize large protein assemblies in mitochondria [96,97,98]. Natural or modified differences between skeletal muscle protein species or protein complexes can be conveniently examined by diagonal non-reducing/reducing 2D-GE following chemical cross-linking [99,100,101]. Following 2D-GE, the next steps typically involve protein spot visualization, using highly sensitive stains, such as Coomassie brilliant blue (CBB) [102], that enable femtomole detection levels of gel-separated and intact proteoforms [103]. Other routinely employed methods for protein spot visualization use silver staining or fluorescent dyes such as SYPRO Ruby or Deep Purple [104,105,106]. 

This is then followed by protein spot abundance analysis and, finally, protein identification by mass spectrometry [107]. In skeletal muscle proteomics, a variety of extremely large myofibrillar and cytoskeletal proteins are difficult to separate by conventional 2D-GE. This includes the giant proteins dystrophin, nebulin, obscurin and titin [108]. To overcome this technical issue, additional analyses can be carried out with a technique complementary to 2D-GE that uses 3–12% gradient 1D-GE in combination with LC-MS [109]. Findings from GeLC-MS/MS can be blended with the results from proteomic surveys employing 2D-GE and can result in more comprehensive insights into the biochemical status of skeletal muscle proteins in the 200–3500 kDa range [110]. Multidimensional protein identification technology (MudPIT) can also be used in conjunction with 2D-GE. Since MudPIT is not based on gel technology for protein separation [111], 2D liquid chromatography prior to MS analysis can add additional proteomic data than might not be as easily assessable by conventional gel electrophoresis [112].

Despite the large number of diverse 2D-GE applications, one significant disadvantage is related to the need to run large numbers of gels, each separating proteins from an individual sample. This limitation was overcome by the development of the difference gel electrophoresis (DIGE) approach [113,114,115]. This technique uses fluorescent cyanine dyes, which are covalently bound to proteins within the samples before the 2D-GE separation begins [116]. The dyes (CyDye Cy2, Cy3 and Cy5) are mass- and charge-matched, but have distinct excitation and emission spectra, allowing for independent signals from the differentially labelled protein populations to be captured [117]. Two different dyes are available: for normal applications, minimal dyes (NHS ester dyes) are used to label lysine residues, and for scarce amounts of sample, saturation dyes (maleimide dyes) are used to label cysteine residues [118,119,120]. Within the DIGE experimental set-up, an internal standard is used (conventionally CyDye Cy2 for minimal dyes and CyDye Cy3 for saturation dyes) [121]. The internal standard can be used to match and normalize the protein quantities across samples [122]. 

Various software packages, including DeCyder, SameSpots and Dymension 3, can be used for the determination of protein spot intensity [123,124,125]. The development of DIGE introduced several advantages for using this research platform in protein analysis and has been modified to also study native protein interactions and post-translational modifications [126,127,128]. Limitations still exist with respect to detecting/resolving low-abundant and hydrophobic proteins, proteins with a molecular mass of <10 kDa or >150 kDa and proteins with an extreme isoelectric point [129]. A significant area where protein separation is based on 2D-GE is the analysis of proteoforms [61], i.e., different molecular forms of a protein product of a single gene that are generated due to alternative mRNA splicing, the activity of more than one promoter per protein-coding gene and/or post-translational proteolysis [130]. As distinct proteoforms may increase or decrease in pathophysiological conditions, the ability to distinguish and quantitate proteoforms is an important consideration when designing an experimental approach [62]. 

#### 2.1.2. Antibody-Based Methodology

Antibodies are perhaps the most frequently used and adapted detectors in biological research, including applications in studies of protein expression, protein interactions, cellular pathways and post-translational modifications (PTMs). Labelled antibodies are a fundamental component of experimental procedures, including immunoblotting [131], immunohistochemistry (IHC) [132], immunofluorescence microscopy (IFM) [133], enzyme-linked immunosorbent assays (ELISA) [134], flow cytometry (FC) [135], fluorescence-activated cell sorting (FACS) [136], mass cytometry (CyTOF) [137] and immunocapture mass spectrometry [138], due to their target specificity and high affinity for specific epitopes.

The most frequently used method to independently verify the mass spectrometric detection of abundance changes in a distinct protein is immunoblotting [139], besides using IHC/IFM methods [140] and enzyme assays [141,142] for the further characterization of proteomic hits. Considerable technological advances have emerged in Western blotting over the years [143]. This has revolved around the introduction of fluorescent-dye conjugated secondary antibodies and the associated ability to multiplex. Using imaging systems to capture the fluorescent signal, researchers can now develop chemiluminescent or fluorescent blots at the bench side. Enhancements to chemiluminescent reagents have made it possible to detect even femtogram amounts of protein, increasing the sensitivity of this approach [139,143]. 

Traditional IHC is commonly used as a technique that assists pathologists in making careful decisions regarding differential diagnosis, disease subtyping and designing personalized treatment plans [144], and plays a key role in the evaluation of skeletal muscle biopsy specimens [145]. IHC and IFM techniques are also used for verification studies in proteomics [133,140]. However, this methodological approach has several limitations, including a high level of inter-observer variability amongst pathologists and the ability to evaluate only one antigen per tissue section. As a result of these limitations, multiplex immunohistochemistry/immunofluorescence (mIHC/IF) technologies, which utilize chromogen-based immuno-detection and antibody stripping chemistry, are now being utilized in both research and clinical settings [146]. The benefits of this platform include increased automation, tissue sparing and cost-effective analysis, as multiple biomarkers can be evaluated on a single formalin-fixed, paraffin-embedded (FFPE) tissue slide [147]. 

Single-plex ELISA tests allow the sensitive and specific detection of various analytes in complex biological samples, such as serum/plasma in clinical and research laboratories, facilitating the diagnosis of diseases and identification of new therapeutic targets [134]. As with recent developments in IHC-associated technologies, there is an increasing requirement for multiplex ELISAs that are capable of obtaining large amounts of data from a limited amount of starting material [148]. Multiplex ELISAs have many advantages over single-plex ELISAs, including increased efficiency, higher throughput and an increase in the number of analytes detected and quantitated [149,150]. Typically, two types of multiplex immunoassays are routinely used; namely, planar and suspension arrays [151]. In planar microarrays, individual capture ligands are immobilized in a microarray format containing potentially several hundred spots and incubated with sample, and then subsequent fluorescent or chemiluminescent signals are detected. In suspension assays, the capture ligands are immobilized onto color-, shape-, or size-coded microspheres. These characteristics are then used to identify the specific analytes that are captured on the bead surface, with quantitation based on the detection of associated reporter molecules, including chemiluminescent or fluorescent signals. Pereira et al. [152] used multiplex ELISAs to investigate oral nutritional supplements enriched with protein, vitamin D and β-hydroxy β-methylbutyrate compared to a control group in serum samples from malnourished sarcopenic older adults. Sixteen biomarkers were found to be significantly changed in response to the supplement, including a decrease in abundance for the inflammation-related ferritin and osteopontin, and an increase in soluble receptors for cytokines, indicating decreased inflammation. To increase the sensitivity of typical multiplex ELISAs, Proximity Extension Assay (PEA) technology has been established. Specific proteins are targeted with a pair of antibodies that are labelled with DNA oligonucleotides, which are hybridized and extended by a DNA polymerase. The DNA barcode that is produced is amplified and quantified using real-time polymerase chain reaction [153].

FC and FACS analysis are partner technologies in the cell analysis process. FC is used for cell analysis and measuring protein expression or co-expression within a heterogeneous population of cells [135]. FACS is used as a cell sorter and enrichment of a subset of cells for subsequent analysis [136]. Recent improvements in FC and FACS have focused on the extension of fluorescent labels (UV and IR range), and novel tandem dyes, allowing for greater multiplexing capabilities. To determine the associations between % circulating osteoprogenitor (COP) cells and sarcopenia, Al Saedi et al. [154] used FC to quantify % COP cells by using selective gating of CD45/osteocalcin (OCN) + cells. Their finding implicates that high levels of % COP cells are associated with better skeletal muscle function when investigating debilitating muscle aging as defined using the Sarcopenia Definitions and Outcomes Consortium (SDOC) criteria [155].

CyTOF, or mass cytometry, uses molecularly tagged antibodies to detect and quantitate specific cellular antigens, allowing for highly multiplexed assays [156]. Heavy-metal isotopic tags, rather than fluorophores, are used to label antibodies, with an increasing number of panels now available for use. Cells are incubated with a mixture of tagged antibodies (non-radioactive heavy metal isotopes) and nebulized, with each droplet containing an individual single cell, and subsequent ionization of the sample [157]. The liberated cloud of ions is subjected to MS-based filtering which selects for the isotope-conjugated probes. In the Time-of-Flight (TOF) chamber, the ions are separated by their mass-to-charge ratio and converted into electrical signals, providing information on the abundance of the specific tagged analytes [137]. CyTOF does suffer from several limitations, including the reduced sensitivity of metal-isotope-tagged antibodies and the longer acquisition times needed when using TOF–MS instruments. However, with these limitations being identified, there is massive scope for advances in these areas that will contribute to increasing the research possibilities of CyTOF in the future of skeletal muscle research [158,159,160]. Recently, Porpiglia et al. [161] used CyTOF to study muscle stem cells (MuSCs) in the aged phenotype and showed high CD47 expression levels, which might be associated with dysfunctional MuSCs, and an impaired regenerative capacity.

In addition, the use of antibodies plays an important role in the affinity precipitation of post-translationally modified peptides prior to MS analysis [162]. Peptides containing a specific modification (such as phosphorylation, acetylation, methylation or ubiquitination), are enriched from protease-digested lysates using an antibody against the specific modification [163,164]. This approach facilitates the identification and quantitation of hundreds to thousands of modified peptides in a single MS run.

#### 2.1.3. Sample Preparation for Proteomic Analysis

The proteomic analysis of skeletal muscle samples is routinely performed with both crude total extracts or subcellular fractions [165]. Subsets of organelles or enriched protein complexes can be isolated by differential centrifugation, density gradient ultracentrifugation, affinity isolation methods or chemical crosslinking approaches [166,167,168]. Optimum protein extraction for subsequent digestion and MS analysis can be carried out by a variety of standardized sample preparation methods [169,170,171]. The filter-aided sample preparation (FASP) technique is ideal for efficient buffer exchange and the removal of MS-incompatible detergents [172]. For designing an optimized proteomic analysis pipeline, it is important to take into account the biological properties of the starting material, such as individual cells, complex tissues or biofluids, and whether a top-down or bottom-up proteomic approach is needed for studying the proteins of interest [58]. Total protein extracts from tough skeletal muscle tissue samples can be conveniently prepared by the FASP method [173]. An alternative methodology for sample preparations is the In-StageTip (iST) technique [174]. In addition, single-pot solid-phase-enhanced sample preparation (SP3) [175] and its variation, named universal solid-phase protein preparation (USP3) [176], can be employed in proteomic applications. If technical complications are encountered with cell or tissue lysis prior to MS analysis [177], these issues can be addressed with recently developed pressure cycling technology (PCT) [178]. In tissue proteomics, the quantification of hydrophobic proteins by MS analysis can be particularly difficult [179,180,181]. Of note, for the proteomic evaluation of large and highly complex protein assemblies, novel high-resolution native MS techniques have been developed [182,183,184]. 

#### 2.1.4. Protein Digestion for Peptide Mass Spectrometry

The controlled and highly reproducible digestion of proteins for the production of a distinct peptide population is an essential requirement for the successful proteomic identification of specific proteoforms. Protein digestion can be carried out by various approaches that differ in the presentation of the proteins of interest in solution, in a gel matrix or on a membrane. One can therefore differentiate between in-solution [185,186], in-gel [187,188] and on-membrane [189,190,191] digestion protocols. The most frequently used protease in MS-based proteomics is trypsin [192], but alternative proteases can be used alone or in combination for protein digestion [193,194,195]. A rapid in-gel digestion protocol was recently designed for GeLC-MS/MS applications [196], which suits the systematic proteomic detection of very large proteins that do not properly move into the second dimension during 2D-GE [110]. An alternative method named BAC-gel dissolution to digest PAGE-resolved objective proteins, BAC-DROP [197], uses the above-described BAC detergent in gel systems, which enables swift solubilization by chemical reduction.

#### 2.1.5. Mass Spectrometric Analysis

The standardized detection of individual proteins in complex mixtures can be routinely performed by MS-based peptide analysis using matrix assisted laser desorption/ionization time-of-flight mass spectrometry (MALDI-TOF) [69,198,199] or liquid chromatography tandem mass spectrometry (LC-MS/MS) [70,71,72,200]. A detailed protocol for LC-MS/MS analysis has been recently published that includes a description of all materials, chemicals, buffers, experimental steps, mass spectrometric parameters and bioinformatic software tools needed for a successful proteomic study [201]. Untargeted quantitative proteomics approaches using mass spectrometry are designed to provide a comprehensive unbiased quantitation of the global proteome using label-free and/or labelling techniques [70,71,72,73]. Label-free quantitation of proteins analyzed by MS uses either integrated peak intensity from the parent-ion mass analysis (MS1) or features from fragment-ion analysis (MS2), including the use of spectral counts. Using next-generation mass spectrometry instruments with high-resolution capabilities and enhanced sensitivity, peak intensity areas from selected parent ions in MS1 can be detected, quantitated and combined with other protein-associated peptides when comparing expression levels between samples [44,45,46]. When using spectral counting, MS2 spectra, generated by peptide fragmentation, are summed with the number of spectra matched to peptides from a specific protein and are then used as a measure of protein abundance. In the field of sarcopenia research, as outlined in more detail below, the label-free MS technique was used by Théron et al. [202] to profile the proteome from *vastus lateralis* muscle samples obtained during surgery from mature and older women. The comparison of protein profiling between these two cohorts identified 35 differentially expressed proteins during skeletal muscle aging, mainly associated with energy metabolism and contractile functionality [202], showing the usefulness of employing label-free MS approaches in sarcopenia research. 

A critical disadvantage of using a label-free approach is that all samples must be measured independently and require significant instrument time in order to achieve a comprehensive analysis of the proteome under investigation. Alternatively, quantitation can be performed using stable heavy isotopes incorporated into proteins by metabolic or chemical labelling protocols [203]. Tandem mass tags (TMT) [204], stable isotope labelling by amino acids in cell culture (SILAC) [205], isobaric tags for relative and absolute quantitation (iTRAQ) [206] and isotope-coded affinity tags (ICAT) [207] are labelling techniques that are routinely used in research studies investigating the proteome under different conditions. TMT labelling, an example of a chemical labelling methodology, is instrumental to quantitative proteomics, especially as the multiplexing approach allows for greater throughput. This enables quantitative analyses with a comprehensive proteome coverage [208].

Each mass-tagging reagent within a set (TMTpro enables multiplexing of up to 16 samples for protein identification and quantitation) has the same nominal mass and chemical structure composed of an amine-reactive NHS-ester group, a spacer arm, and an MS/MS reporter. The intensity of the unique MS/MS reporter ions (different *m*/*z*), detected using LC-MS/MS, is used to determine the amount that each peptide from the labelled samples contributes to the selected parent mass, facilitating relative quantitation. TMT-based proteomics has the advantage of higher quantitative accuracy, fewer missing quantitative values among samples, and reduced sample run times on MS instruments. TMT probes have been used in aging research to quantitate the proteome from young versus old rats [209]. The comparative analysis of slow-twitching *soleus* muscles versus fast-twitch *extensor digitorum longus* muscles revealed 78 and 174 proteins being differentially expressed during aging, respectively, and were shown to be generally associated with energy metabolism, oxidative stress, detoxification and transport [209].

SILAC is a quantitative proteomic approach using metabolic labels, which allows the comparison of cultured cells (lysates/secretome) under different conditions [204]. Using this approach, identification and quantitation of thousands of proteins can be performed in a single experiment by combining differently labelled samples prior to analysis by LC-MS/MS [210]. A standard SILAC experiment can be used to compare two or three samples by labelling with a light label (standard media), medium label (media containing ^2^H_4_-lysine and ^13^C_6_-arginine) and a heavy label (media containing ^15^N_2_^13^C_6_-lysine and ^15^N_4_^13^C_6_-arginine) [211]. The complete incorporation of heavy amino acids during protein turnover, in combination with the use of trypsin as the digestive enzyme, means that peptides from the differentially labelled samples can be accurately quantified relative to each other, based on the defined mass difference between the samples [212]. 

In skeletal muscle proteomics, SILAC was used to study differentiation, fiber damage and fiber typing [213,214,215]. An interesting application of SILAC in combination with an immunoaffinity protocol was the investigation of muscular atrophy in mice that were fed a SILAC diet containing ^13^C_6_-lysine for 4, 7 or 11 days when comparing denervation-induced changes after sciatic nerve section in the *gastrocnemius* muscle as compared to control samples [216]. Ubiquitin remnant peptides (K-ε-GG) were profiled by immunoaffinity enrichment, with results showing that >2100 diglycine remnants were identified, providing an insight into the ubiquitination process during muscular atrophy [216].

Dynamic proteome profiling (DPP) with a deuterium label can be employed to determine time-dependent changes in peptide mass isotopomer abundances [217]. The DPP technique was recently applied to study the relative abundance and fractional synthesis rate of proteins in human muscle biopsy specimens [218], and during C2C12 myoblast differentiation [219] and cellular aging [220]. As listed below, a study by Murphy et al. [221] of obese and healthy men of old age, who underwent resistance training and caloric restriction, determined the amount of newly synthesized skeletal muscle proteins via deuterated water labeling.

Importantly, MS analyses combined with artificial intelligence (AI) are increasing the potential for research and analysis of proteins in the field of proteomics [222]. The MS approach has proven to be a pillar for quantitative studies in addition to the identification of PTMs. Higher-plexing labelling reagents, in combination with advanced data acquisition protocols using the next generation of instruments, provide data on hundreds of thousands of protein isoforms in large sample cohorts. As datasets are becoming more all-encompassing, the use of AI, along with Machine Learning (ML) and Deep Learning (DL) algorithms, will become common features for analyzing the complex spectral data to identify pathophysiological patterns for actionable biology.

#### 2.1.6. Data Acquisition by Mass Spectrometry

Data acquisition by mass spectrometry can be performed using data-dependent acquisition (DDA) [223], data-independent acquisition (DIA) [224] and targeted data acquisition (TDA) [225]. The DDA analysis mode involves using the MS instrument to generate a full-scan mass spectra (MS1), where the N most intense peptide ions (i.e., top 15) are selected and MS/MS spectra acquired. This approach generates thousands of MS/MS spectra that can be used for protein identification and subsequent quantitation. However, as the most abundant peptide ions are selected in the full scan, lower abundant peptide ions are repeatedly excluded from selection, even when using filtering criteria such as dynamic exclusion. 

The DIA analysis mode involves using the MS instrument to direct the analysis on a narrow mass window of precursors and acquiring MS/MS data from all precursors detected within that window [226]. By stepping across the defined mass range using specific mass windows, collected MS/MS data will be acquired from all detected precursors. This strategy then uses highly specific fragment ion maps in a spectral library for qualitatively and quantitatively analyzing DIA data sets [227]. Sequential window acquisition of all theoretical mass spectra (SWATH-MS), as described by Gillet et al. [228], is a common method to generate DIA data by dividing the mass range into small mass windows.

The verification phase of many proteomics investigations centers on confirming that the abundances of target peptides are significantly different between sample cohorts by using MS-derived quantitative measurements. Selected/Multiple-Reaction Monitoring (SRM/MRM) or Parallel-Reaction Monitoring (PRM) are examples of approaches that can be utilized, where precursor peptide ions are measured in predefined *m*/*z* and retention time [229,230,231]. Stable-isotope-labelled, synthetic peptides are often spiked into the samples of interest, a process that increases the overall accuracy of target peptide quantitation. 

#### 2.1.7. Single-Cell Proteomics

Within the last decade, single-cell RNA sequencing (scRNA-seq) has come to the fore as an informative approach to decode tissue composition at the single-cell level and to provide important mechanistic data about pathophysiological associated networks [232]. Protein abundance in single cells is often deduced from complementary analysis platforms (scRNA-seq), as the ability to quantitate the proteome at a single-cell level has remained challenging [233]. Initial approaches for quantitating proteins in single cells relied on antibodies. Hence, these methods depend heavily on the availability of high-quality antibodies, therefore limiting their impact in the analysis of many antigens [234]. However, the use of MS/MS combined with LC-based separation is gaining traction with respect to its application in the analysis of the single-cell proteome [235].

Notable breakthroughs in this area include the use of isobaric labelling for single-cell proteomics, called Single Cell ProtEomics by Mass Spectrometry (SCoPE-MS) [236], and the second-generation protocol called Single Cell ProtEomics (SCoPE2) [237]. Such protocols permit cells from heterogeneous populations to be adapted into single-cell suspensions by FACS or CellenONE [238,239]. CellenONE is a precision dispensing technology combined with advanced image processing that delivers real-time and high-accuracy single cell isolation and dispensing. The isolated single cells are lysed, proteins digested, and the resultant peptides labelled with TMTs [240]. The different steps of this protocol can be automated, allowing for reproducibility and scalability. Labelled peptides are mixed and analyzed by MS/MS combined with LC [241]. 

Label-free analysis of individual cells does not require the use of TMTs, but their throughput is lower than that of the labelling approach [242]. The use of TMTs and the ability to multiplex ultimately increase the amount of peptides detected and quantitated by MS, which is particularly important when analyzing small-diameter mono-nucleated cell populations. However, the analysis of skeletal muscle fibers has some advantages, given that these types of fibers are multi-nucleated single cells and relatively bulky compared to other cell types. Individual muscle fibers contain on average a few micrograms of protein, and their isolation by dissection is more straightforward than having to use FACS or CellenONE approaches. A recent manuscript by Murgia et al. [243] demonstrated the utility of single-cell proteomics when comparing the proteome of type 2X fibers to that of type 1 and 2A fibers in young individuals. Their dataset contained more than 3800 proteins detected by single-fiber proteomics, with approximately 10% of the identified proteins displaying a statistically significant difference among the fiber types investigated. This approach has the potential to increase our understanding of musculoskeletal tissue development and disease within individual muscle fibers [244]. The application of single-cell proteomics in muscle aging is discussed below. Importantly, nanotechnology is increasingly used for optimum sample preparation in single-cell proteomics, as discussed by Arias-Hidalgo et al. [245].

#### 2.1.8. Aptamer-Based Proteomics

Other proteomic-based platforms that are growing in popularity and number include aptamer-based approaches [246]. Aptamers are single strands of oligonucleotides (either ssDNA or ssRNA) that bind with high specificity and high affinity to preselected proteins [247]. The range of the preselected protein panels is ever increasing, with one of the leading aptamer-based proteomics platforms, SomaLogic, offering different protein panels ranging from 1300 to over 7000 targets in as little as 55 mL of plasma or serum. Hathout et al. [248] recently used the SomaLogic platform to identify 108 elevated and 70 decreased proteins in dystrophic patients who were not yet treated with glucocorticoids compared to age-matched healthy controls. High-throughput multiplexing techniques can be combined with TMT technology to detect serum biomarkers that have been released from damaged skeletal muscle fibers [249].

### 2.2. Proteomic Profiling of Fiber-Type Specification in Skeletal Muscles

Most individual skeletal muscles consist of a distinct mixture of fast-twitching, slow-twitching and hybrid fibers [250,251,252], and this fiber-type composition can undergo substantial alterations during progressive muscle wasting [50,51,253]. Fiber-type specification has traditionally been determined by histological, histochemical and immunohistological staining procedures [132,254,255]. Recently, Kallabis et al. [215] described a novel high-throughput proteomic workflow for myosin isoform profiling in single muscle fibers based on the usage of a capillary LC-MS gradient in a 96-well format. This is an excellent improvement of the fiber-type-specific screening of the skeletal muscle proteome. Over the last two decades, the steady improvement of protein separation methodology and mass spectrometric detection efficiency, in combination with enormous advances in bioinformatics, has resulted in the greatly enhanced coverage of the skeletal muscle protein constituents [65,66,67,256]. 

A large number of proteomic markers are now available for the comprehensive profiling of subcellular fractions from skeletal muscles [257]. Over 10,000 protein species belonging to the core proteome of human and animal skeletal muscles have been identified and characterized by mass spectrometry [258,259,260,261,262,263]. The proteomic profiling of differing skeletal muscles with specific fiber-type distribution patterns has especially focused on human *vastus lateralis*, *deltoideus* and *trapezius* muscles [264,265,266] and mouse *gastrocnemius*, *soleus* and diaphragm muscles [267,268,269]. Comparative MS-based studies of mouse *extensor digitorum longus* and *soleus* muscles [270,271,272], and tissue extracts from rodent *gastrocnemius*, *extensor digitorum longus*, *tibialis anterior* and *soleus* muscles [273,274,275,276], have given comprehensive insights into the biochemical complexity of fiber-type-specific protein expression patterns using single-fiber proteomics [277]. The study by Eggers et al. [276] utilized immunolabeling of individual skeletal muscle fibers with antibodies to specific myosin heavy chain isoforms followed by laser micro-dissection and MS analysis. The detailed biochemical characterization of mouse muscle fibers by single-cell proteomics revealed an in-depth profile of fiber-type-specific protein expression levels [276].

### 2.3. Composition of the Acto-Myosin Apparatus and Its Proteomic Profile

Skeletal muscle fibers are highly specialized cellular structures for the generation of force and movement [278]. The sarcomeric components of the acto-myosin apparatus [279] provide the molecular machinery for coordinated filament sliding during skeletal muscle contractions [280]. Contractile proteins exist in a large number of isoforms [281] and can be divided into groups of proteins that are mostly located in the thick myosin-containing filament [282], the thin actin-containing filament [283], the M-line [284] and the Z-disk [285], as well as auxiliary filamentous structures [286]. The sarcomere units have extensive intrinsic connections [287] and are embedded in the overall muscle structure by an extensive cytoskeletal system linking them to organelles for energy supply and signaling mechanisms, and to the costamers for force transmission [288]. Figure 1 provides an overview of the contractile acto-myosin apparatus within the sarcomeric structure of skeletal muscles.

Slow versus fast isoforms of key sarcomeric proteins are displayed in Figure 2 below. The abbreviations of specific muscle protein isoforms are listed at the end of the manuscript, and are used throughout the text, Tables and Figures. The names of genes are exclusively listed in italics to avoid confusion with abbreviated protein names. 

In the thick filaments of skeletal muscles [282,287], the hexameric composition of the major motor protein myosin consists of two myosin heavy chains (MyHCs) [289] and four myosin light chains (MLCs) [290], which can be further subdivided into two phosphorylatable regulatory light chains and two non-phosphorylatable alkali light chains [291]. The main MyHC isoforms in human skeletal muscle are the slow type I isoform MyHC-1 (*MYH7* gene), the fast type IIa isoform MyHC-2a (*MYH2* gene) and the fast type IIx isoform MyHC-2x (*MYH1* gene) [250,252,292]. Another fast isoform of type IIb is named MyHC-2b (*MYH4* gene), and is present at high concentration only in small mammals, such as mice, rats and rabbits [110]. Type IIb fibers with high levels of MyHC-2b are extremely fast-contracting and quickly fatigable units that are usually not found in mature human skeletal muscles [251]. In addition, MyHC-emb (*MYH3* gene), MyHC-neo (*MYH8* gene) and six other MyHC isoforms, encoded by the genes *MYH6*, *MYH7B*, *MYH13*, *MYH14*, *MYH15* and *MYH16*, respectively, exist in embryonic/fetal muscles [293] and specialized adult muscles, including masticatory, extraocular and laryngeal muscles, as well as muscle spindles [294,295,296]. The recent proteomic profiling of extraocular muscles has also detected, besides the long-established MyHC-13 isoform, MyHC-14 and MyHC-15 being present in these highly specialized and mostly fast-twitching muscles [297]. The slow and fast isoforms of MLC proteins are represented by slow/cardiac regulatory light chain MLC-2s (*MYL2* gene), fast regulatory light chain MLC-2f (*MYL11* gene; with the previous HGNC gene symbol *MYLPF*), slow essential light chain MLC-1s (with isoform MLC-1sb encoded by the *MYL3* gene; and MLC-1sa encoded by the *MYL6B* gene) and fast essential light chain MLC-1/3f (with MLC-1 and MLC-3 being splicing products of the *MYL1* gene) [289,291,298]. Myosin-binding proteins (MYBP) [299] are located at the thick filament interface and are present as slow and fast isoforms, i.e., MYBP-C1 (slow myosin-binding protein C1; encoded by the *MYBPC1* gene) and MYBP-C2 (fast myosin-binding protein C2; encoded by the *MYBPC2* gene) [300,301,302].

In the thin filament [303], the basic units that form helical actin (ACT) filaments are alpha-actin-1 monomers of the skeletal muscle ACTA type (*ACTA1* gene) or the cardiac muscle ACTC type (*ACTC1* gene) [304,305]. The Ca^2+^-dependent process of regulating interactions between the MyHC heads and ACT filaments is provided by tropomyosin (TPM) and the troponin (TN) complex [306,307]. Sarcomeric TPM molecules are alpha-1-tropomyosin (TPM-1; encoded by the *TPM1* gene), slow beta-tropomyosin (TPM-2; encoded by the *TPM2* gene) and muscle-type alpha-3-tropomyosin (TPM-3; encoded by the *TPM3* gene) [308,309]. The alpha-4-tropomyosin isoform named TPM-4 (*TPM4* gene) is a non-sarcomeric cytoskeletal component [310]. The TN complex consists of the Ca^2+^-binding subunit TNC, the TPM-interaction subunit TNT and the inhibitory subunit TNI [311]. All three subunits exist in fast and slow isoforms and exist in various combinations in matured skeletal muscles [312]. This includes TNC-1, the slow/cardiac troponin TnC isoform (*TNNC1* gene), TNC-2, the skeletal muscle troponin TnC isoform (*TNNC2* gene), TNT-1, the slow muscle troponin TnT isoform (*TNNT1* gene), TNT-3, the fast muscle troponin TnT isoform (*TNNT3* gene), TNI-1, the slow muscle troponin TnI isoform (*TNNI1* gene) and TNI-2, the fast muscle troponin TnI isoform (*TNNI2* gene) [313]. In addition, the cardiac isoforms TNNI3 and TNNT2 have been found in aged and denervated skeletal muscles [314]. 

The Z-disk contains a large number of proteins, including filamin-C (FLNC; *FLNC* gene), telethonin/titin-cap protein (TCAP; *TCAP* gene) and alpha-actinin (ACTN) with its closely associated binding protein myozenin (MYOZ) [315,316]. They are excellent subcellular markers of this crucial sarcomeric structure [285]. The major ACTN proteins found in the Z-disc are the alpha-actinin-2 isoform ACTN-2 (*ACTN2* gene) and the alpha-actinin-3 isoform ACTN-3 (*ACTN3* gene) [317]. Interestingly, the *ACTN3* genotype appears to be linked to the maintenance of bone and skeletal muscle mass during aging [318]. MYOZ isoforms that are present in skeletal muscles are MYOZ-1 (myozenin-1; previously named FATZ-1; *MYOZ1* gene), MYOZ-2 (myozenin-2; *MYOZ2* gene) and MYOZ-3 (myozenin-3; *MYOZ3* gene) [319,320]. Excellent marker proteins of the M-line structure of the sarcomere [321] are the myomesin (MYOM) proteins MYOM1 (myomesin-1; *MYOM1* gene) and MYOM-2 (myomesin-2; *MYOM2* gene) [322,323], as well as obscurin (OBSCN; *OBSCN* genes) [324,325]. The M-line-associated obscurin molecule belongs to the class of giant muscle proteins [108]. Two other major sarcomeric components are also characterized by extremely high molecular masses, i.e., the actin-binding protein nebulin (NEB; *NEB* gene) of the thin filament [326,327] and the half-sarcomere spanning component titin (TNN; *TTN* gene) [328,329] with multifunctional roles in lattice order, filament interactions and the excitation–contraction–relaxation cycle [330,331]. Closely linked to titin is the muscle ankyrin repeat protein MARP (*ANKRD2* gene) [332].

As a representative example of how proteomics can be employed to routinely detect and characterize a large number of specific isoforms of contractile proteins, Table 1 lists the mass spectrometric identification of major sarcomeric proteins that are associated with the thick myosin filament, thin actin filament, the titin filament, the Z-disc and the M-line in diaphragm muscle [269,333]. The information presented includes the protein names and abbreviations of particular isoforms, their accession number, the name of the coding gene, percentage of sequence coverage, number of peptides and calculated molecular mass. As listed in Table 1, diaphragm muscles are characterized by the presence of MyHC-1, MyHC-2x, MyHC-2b, MyHC-8, MLC-1/3, MLC-2 and MLC-3 in the thick filament, and muscle-type ACTA and various slow and fast isoforms of TPM, TNC, TNI and TNT in the thin filament. Abundant components in the Z-disc were established to include FLNC, TCAP, ACTN isoforms and MYOZ, and the M-line was shown to contain MYOM and OBSCN. The muscle protein that was recognized by the highest number of peptides is represented by the giant sarcomeric protein TTN [108]. A closely linked component of the titin filament was identified as the muscle ankyrin repeat protein MARP.

## 3. Proteomics of Age-Related Muscle Wasting

### 3.1. Pathobiological Hallmarks of Sarcopenia of Old Age

Skeletal muscle aging can be considered a fundamental biological process that occurs in all humans of advanced age [334]. However, individual muscles in the same body age differently [28,335] and considerable inter-individual differences exist in the extent and time course of muscle tissue loss and decline in contractile strength [9,336]. Importantly, skeletal muscle degeneration can be accompanied by progressive deterioration of myocardial functions in the elderly, causing serious medical complications due to cardio-sarcopenia syndrome [337]. Although sarcopenia of old age is due to multi-factorial mechanisms, it is most likely that neurological changes during aging play a key role in the initiation of muscular atrophy. The loss of spinal motor neurons appears to be associated with the initial decline in the proper innervation of voluntary muscles. The damage of the neuronal systems is exacerbated by a diminished capacity for reinnervation or patterns of faulty reinnervation [338]. The tendency of fast-to-slow muscle fiber-type transitions in a large number of aged human muscles was shown to be linked to a higher susceptibility of faster-contracting fibers to muscular atrophy [50,51,339]. This higher vulnerability of faster-twitching and mostly glycolytic fibers under atrophic conditions is closely related to specific signaling pathways involving peroxisome proliferator-activated receptor gamma coactivator PGC1-alpha and transforming growth factor TGF-beta [340].

Epidemiological studies of sarcopenia, assessed by both cross-sectional and longitudinal investigations, indicate that everyday life of a large proportion of the population over 75 years of age is impaired by a certain degree of physical frailty and impaired skeletal muscle functioning [341,342,343,344,345,346,347]. Worsening cofactors of age-related muscle wasting include sarcopenia-independent chronic diseases and their extensive pharmacological therapy, as well as chronic low-grade inflammation, insulin resistance, poor nutrition, extended bedrest and the lack of appropriate physical activity levels [9,10,14,348,349]. Thus, to counteract the age-dependent decline in skeletal muscle performance, optimized rehabilitation [350] and appropriate physical exercise regimes, such as moderate resistance exercises [351,352,353], are crucial to minimize oxidative stress and inflammation in sarcopenia [354,355]. Since older adults exhibit a higher rate of protein turnover [356], and an apparent imbalance between accelerated muscle protein breakdown and impaired levels of protein re-synthesis exists in aged muscles [9,10,11], the resulting reduced levels of contractile components in older individuals should be addressed by avoiding a poor diet quality [357,358,359] and instead provide an adequate intake of high-quality protein in the elderly [360,361,362,363].

Recent publications have critically examined the diverse and multi-factorial aspects of aging and sarcopenia, including senescence-related changes linked to abnormal metabolic pathways [364], mitochondrial dysfunctions [365,366,367,368,369,370,371], the role of reactive oxygen species and disrupted redox signaling [372,373,374,375], abnormal calcium handling [376], functional changes in neuromuscular transmission [377], altered myokine and myomitokine signaling [369,378], the role of miRNAs in the decline of proteostasis [379,380], anabolic resistance and impaired muscle protein metabolism [381,382,383], adipocyte crosstalk in aged skeletal muscle and sarcopenic obesity [384,385], immune system alterations, chronic inflammation and immune–metabolic dysfunction associated with oxidative stress [33,366,386,387,388], the role of telomere length during aged fiber regeneration [389], the interplay between sarcopenia, frailty and cognitive impairments in the elderly [390], cardio-sarcopenia syndrome [337] and the influence of nutrition on the aging phenotype [391]. The finding that the satellite cell pool is preferentially affected in fast type II fibers in the elderly [392] has established the idea that stem cell exhaustion is majorly involved in sarcopenia and possibly even facilitates age-associated fast-to-slow transitions [393]. Thus, the reduction in muscle-specific stem cells appears to play a key role in the impaired regenerative capacity of aged fibers [394,395,396]. This phenomenon underlines the enormous complexity of the molecular and cellular mechanisms that are associated with skeletal muscle aging.

### 3.2. Proteomics of Aged Skeletal Muscle

Biomarker discoveries using omics-type surveys are crucial to improve the monitoring of impaired physiological functioning, altered energy metabolism and chronic inflammation in aged muscle [397], and to advance the diagnosis, prognosis and therapeutic monitoring of frailty syndrome and sarcopenia in the aging population [398,399,400], whereby proteomics plays a key role in detecting and characterizing novel marker candidates [48,49]. In the context of aging and alterations in contractile proteins, human skeletal muscles were extensively studied using both top-down/gel-based approaches versus bottom-up/peptide-centric analyses [202,221,401,402,403,404,405,406,407,408,409,410,411,412,413,414]. Changes in particular isoforms of skeletal muscle proteins during the aging process can give detailed insights into molecular and cellular mechanisms that underlie sarcopenia of old age. Although individual skeletal muscles exhibit differing degrees of susceptibility to aging-induced muscular atrophy [28], proteomics has confirmed the previous findings from biochemical, cell-biological and histological studies that suggest a general trend of fast-to-slow transitions in senescent muscles [50,415,416] and concomitant alterations in glycolytic and mitochondrial pathways [39,417]. This includes a stepwise transition from faster isoforms of MyHC, MLC, ACT, TPM, TNC, TNI and TNT to their slower counterparts. Of note, the recent proteomic profiling of single fibers from human *vastus lateralis* muscle of young adults has given a comprehensive overview of fiber-related differences in protein isoform expression patterns [243]. These types of proteomic catalogs can be highly useful as reference databanks for studying proteome-wide changes during aging.

Table 2 lists major MS-based investigations with a bioanalytical focus on protein changes in contractile proteins during human skeletal muscle aging [202,221,401,402,403,404,405,406,407,408,409,410,411,412,413,414]. The listings of individual proteomic investigations summarize the analyzed muscle specimens, the age range of samples, the bioanalytical approach and the detected proteome-wide alterations with a focus on the contractile apparatus. Since considerable physiological and biochemical differences exist between untrained versus trained skeletal muscles [68,418,419,420], contractile fiber aging has also been studied in select master athletes [421,422] in addition to the below-listed studies on neuromuscular changes in the general and mostly untrained population. Major proteomics surveys of aged human muscles that did not focus on the contractile apparatus include investigations into the role of mitochondrial abnormalities [423] and molecular chaperones [424], as well as metabolic changes due to oxidatively modified proteins in satellite cells [425].

Top-down proteomics using routine 2D-GE or fluorescent 2D-DIGE is an ideal bioanalytical approach for the efficient separation of contractile proteins below 150 kDa [122], such as fast and slow isoforms of MLC, TPM, TNC, TNI, TNT and ACT [52,66]. Human skeletal muscles usually contain a mixture of slow-twitching fibers, which are characterized by high levels of oxidative metabolism, and faster-twitching fibers with glycolytic-oxidative or mostly glycolytic metabolism [250], in addition to various hybrid fibers [426]. Mass spectrometric analyses of separated 2D spots clearly confirmed shifts from fast protein isoforms to their slower protein counterparts [401,402,403,404,405,409,413], which agrees with the general tendency of fast-to-slow transitions during skeletal muscle aging [50,51,253,339]. These findings could be complemented with bottom-up strategies and LC-MS/MS analyses to study contractile proteins of higher molecular mass, such as MyHC and TNN [202,221,406,407,408,410,411,412]. In analogy to shifts towards faster isoforms of TPM and subunits of the TN complex, LC-based studies confirmed transitions from MyHC-2 isoforms towards MyHC-1. In addition, the application of iTRAQ demonstrated decreases in ACTA and FLNC [414]. Single-muscle-fiber proteomics showed differential effects on fast versus slow fibers based on the mass spectrometric detection of MyHC-1, MyHC-2a and MyHC-2x distribution patterns [407]. Overall, the findings from the proteomic analysis of aged human muscles, focusing mostly on the *vastus lateralis* muscle, agree with the higher susceptibility of fast fibers to atrophic changes [202,221,401,402,403,404,405,406,407,408,409,410,411,412,413,414] and support the cell biological concept of fast-twitching fibers being affected prior to slower fiber population during skeletal muscle aging [416]. 

In analogy to the above-listed studies on human skeletal muscles, the analysis of various animal muscles revealed similar tendencies of fast-to-slow transitions during fiber aging. For example, the mass spectrometric profiling of the aging *vastus lateralis* muscle from African green vervet monkeys (*Chlorocebus aethiops sabaeus*) confirmed decreases in fast MyHC isoforms during age-related muscular atrophy [427]. Most aged animal studies were carried out with small rodents [39,428,429,430]. Interesting new model systems used in aging research are *Drosophila*, zebrafish and nematodes [431,432,433,434]. Following initial optimization experiments [435,436], several proteomic investigations of animal models focused on the analysis of mitochondria [437,438,439,440,441,442], the matrisome [443,444], the cellular stress response [445], calpain-interacting proteins [446] and key post-translational modifications [447], such as glycosylation [448], phosphorylation [449], carbonylation [450] and nitration [451,452,453,454,455] in aged muscles. General alterations in the senescent skeletal muscle proteome, including abundance changes and isoform switching of contractile proteins, were examined in a large number of MS-based surveys using both the well-established rat model of sarcopenia [209,405,456,457,458,459,460,461,462,463,464,465,466,467,468,469,470,471,472,473] and aging mouse muscles [474,475,476,477,478,479,480,481,482,483,484,485,486]. The gel-based analysis of aging rat *gastrocnemius* muscle clearly identified decreases in MyHC-2b, MLC-2f and TPM-1 as compared to increases in MyHC-1, MLC-2s and ACTC [456,457,458,459,460,461,462,463,464]. In particular, MLC-2s appears to be majorly affected both in its abundance and phosphorylation pattern during skeletal muscle aging [461,466], making it an excellent biomarker candidate of fiber-type switching.

As illustrated in the representative findings on abundance changes in Ca^2+^-regulatory components and contractile proteins in Figure 3, MS-based proteomics is an excellent bioanalytical tool to establish decreases in important Ca^2+^-handling proteins that are involved in cellular signaling pathways and the regulation of excitation–contraction coupling. This includes subunits of the dihydropyridine receptor L-type Ca^2+^-channel of the transverse tubules, the ryanodine receptor Ca^2+^-release channel of the triad junction, the luminal Ca^2+^-binding protein calsequestrin of the terminal cisternae region within the sarcoplasmic reticulum and the structural protein triadin [487,488,489]. Thus, a key mechanism involved in skeletal muscle aging appears to be a certain degree of pathophysiological uncoupling between sarcolemmal excitation and the initiation of acto-myosin sliding that mediates fiber contraction [376,490,491,492], combined with a reduced association between Ca^2+^-release units and aged mitochondria [493]. Disturbed Ca^2+^-homeostasis may be involved in altered myocyte signaling in the context of fiber-type specification, which is supported by proteomic data that indicate a general tendency of fast-to-slow transitions at the level of isoform switching of contractile proteins [50].

## 4. Age-Related Muscular Atrophy, Biomarker Discovery and Therapeutic Approaches

### 4.1. Mechanisms of Age-Related Muscular Atrophy

Research over the last few decades has clearly established that the molecular and cellular mechanisms of aging are highly complex [30,31,32] and specifically affect the skeletal musculature [9,10,11]. The multi-factorial processes that are associated with age-related muscular atrophy and sarcopenia of old age include: Progressive neurodegeneration: loss of neuromuscular junction integrity; degeneration of motor neurons and resulting denervation; faulty patterns of reinnervation; loss of entire motor units;Excitation–contraction uncoupling at the level of the transverse tubules, triad junction and sarcoplasmic reticulum;Impaired calcium homeostasis;Abnormal mitochondrial functioning;Fast-to-slow transitions due to increased susceptibility of fast fibers to atrophy;Tendency of bioenergetic glycolytic-to-oxidative shifting;Increased cellular stress due to proteotoxic abnormalities;Abnormal protein turnover and synthesis causing dysregulated proteostasis;Hormonal imbalance and disturbed cellular signaling;Visceral obesity causing abnormal muscle-fat-axis signaling;Metabolic syndrome and insulin resistance;Increased levels of reactive myofibrosis triggering loss of fiber elasticity;Chronic low-level sterile inflammation;Reduced regenerative capacity due to satellite cell exhaustion;Epigenetic changes.

Figure 4 summarizes crucial aspects of muscle aging, including the preferential susceptibility of fast type II fibers to age-related degeneration, which causes a general shift to slower-twitching fiber populations in most senescent skeletal muscles.

### 4.2. Biomarker Discovery for the Improved Evaluation of Sarcopenia of Old Age

In order to improve the differential diagnosis of pre-, mild or severe sarcopenia [9,10,11], the establishment of reliable and robust biomarkers of frailty and skeletal muscle wasting is crucial [29,49]. Suitable markers can be measured by physical performance assessments [25,26,27], imaging technology [22,28] and/or biochemical assays [494]. A novel imaging marker system is the ultrasound sarcopenic index (USI), which can determine the loss of skeletal muscle mass in association with sarcopenia in a practical and relatively inexpensive way [495]. Ideally, abundance changes in protein biomarkers of sarcopenia should be easily measurable with high levels of specificity and sensitivity [496,497], as well as not being majorly affected by gender, ethnicity, co-morbidities, exposure to pharmacological agents and unrelated therapeutic treatments [498]. To avoid potential complications due to elaborate tissue biopsy procedures, the development of non-invasive disease indicators is favorable [499]. A recent meta-analysis of proteomic studies by Stalmach et al. [500], using a gene ontology-driven approach, suggests that it is advantageous to integrate MS data sets from both muscle tissue samples and suitable biofluids to gain more comprehensive insights into atrophying changes in the human skeletal muscle proteome. 

This gives non-invasive biomarker investigations of biological fluids, such as serum, saliva or urine, a central role in aging research [501,502,503]. The serum of both older humans suffering from sarcopenia [504,505,506,507] and senescent mice [508] were shown to exhibit differential changes in common markers that are associated with inflammation, remodeling of the extracellular matrix and mitochondrial functions [398]. This suggests the potential usage of pro-inflammatory cytokines, growth factors, differentiation factors and leaked mitochondrial proteins as suitable biofluid markers to evaluate the degree of skeletal muscle aging [400]. The regulatory factor myostatin and insulin growth factor IGF-1 show considerable potential to be useful as gender-specific markers of low skeletal muscle mass and frailty [509]. Ideally, proteomic findings are correlated to the results from systematic transcriptomic and metabolomic studies of sarcopenia [510,511,512].

Promising biofluid protein markers of sarcopenia are the carboxy-terminal fragment of agrin (CAF) [513,514,515,516] and the brain-derived neurotrophic factor [517,518,519]. The proteoglycan agrin is closely associated with the sarcolemmal dystrophin/utrophin-glycoprotein complex that is involved in the cytoskeletal stabilization of the neuromuscular junction [520]. The loss of neuromuscular junction integrity appears to play a key role in muscular atrophy [377,521] including sarcopenia of old age [522]. The activity of the synapse-specific protease neuro-trypsin [523], and agrin cleavage, are clearly related to the age-dependent degeneration of the neuromuscular junction [513]. The remodeling of aged motor units in turn is linked to the preferential denervation of fast-twitching and mostly glycolytic type II fibers, and faulty patterns of reinnervation by smaller motor neurons that establish slower-contracting type I motor units [524,525,526]. 

The age-related fiber-type shifting and accompanying changes in MyHC isoforms [527] can only generate lower maximum force levels in senescent skeletal muscles as compared to young and adult muscle systems. These alterations in the overall composition of motor units probably plays a central role in the gradual loss of skeletal muscle strength during aging [528]. This makes circulating CAF a potential biofluid biomarker of motor unit changes in sarcopenia, in conjunction with fast-to-slow fiber-type shifting in aged muscle tissues, as outlined in Figure 5.

### 4.3. Therapeutic Approaches to Counteract Age-Related Muscular Atrophy

Aging-associated processes lead to a general decline of health status, a higher risk of disease and drastically reduced physical fitness. It is crucial to take multi-system derangements into consideration when designing novel therapeutic approaches to treat individual age-related ailments, such as sarcopenia. Frailty syndrome can result in a diminished quality of life and even loss of independence in the case of severe and chronic muscle wasting. General recommendations to support healthy aging include the positive influence of a healthy and balanced diet, sufficient sleep, regular relaxation, proper physical exercise, calm breathing patterns, regular social interactions and a positive view of life [529,530,531,532]. Thus, promoting a healthy lifestyle should include countermeasures against sarcopenia of old age to avoid the premature loss of physical strength and skeletal muscle mass. However, a crucial issue for the elderly is proper access to advanced strength training equipment and the realistic implementation of health-promoting support structures, especially during pandemics. During the current COVID-19 crisis, the aged population has only limited access to gyms, parks, recreational facilities and rehabilitation services, causing long-term negative effects on muscle health [533,534,535], and this situation has to be urgently addressed to promote healthy aging. In addition, the treatment of acute sarcopenia in patients with or without COVID-19 infection has been complicated by the restricted access to health services during the pandemic [536,537,538,539], and the increased application of mechanical ventilation and complications during ventilator weaning has caused considerable side effects, including skeletal muscle wasting [540].

Therapeutic approaches to attenuate the impact of age-related skeletal muscle degeneration include non-pharmacological interventions, such as lifestyle changes that incorporate regular and appropriate resistance training [351,352,353,541,542], and optimized dietary considerations, including a protein-rich diet and the frequent ingestion of small portions of high-quality food [358,359,360]. Nutritional combinations of Vitamin D, leucine-enriched protein supplements and whey protein were shown to have some effects on building skeletal muscle mass and improve physical functionality of the neuromuscular system [543,544,545,546]. The combination of mixed types of regular physical exercise with a balanced diet and nutritional supplementation appears to be the most suitable multi-component intervention strategy to minimize the effects of sarcopenia and avoid mobility disability in older adults [547,548]. A protein-rich diet, combined with high levels of physical activity, should both stimulate muscle protein synthesis and thus prevent impaired proteostasis in senescent fibers, and have generally positive effects on metabolism, bioenergetics and hormonal balance. At advanced age, combining a low-intensity form of home-based resistance exercise with proper nutrition and a multi-ingredient supplementation seems to be the most effective way to treat sarcopenia.

Regular exercise has a profound effect on the skeletal muscle proteome [68,354,419] and muscle fiber-type diversification [549,550,551]. In particular, resistance exercise aimed at improving the contractile strength of aged skeletal muscles is generally associated with alterations in myofiber size, muscle re-innervation, fiber-type-specific myonuclear adaptations, mitochondrial remodeling and fiber-type shifting [552,553,554,555,556,557]. Since the age-related loss of skeletal muscle mass is mostly due to a drastic reduction in the size of fast-twitching type II fibers [50,51,339,416,558], it is encouraging that resistance exercise specifically results in the hypertrophy of type II muscle fibers, although it does not appear to affect patterns of fiber-type grouping in aged muscles [559]. Distinct changes in MyHC isoform expression patterns are usually exemplified by reduced MyHC-1 and increased MyHC-2x levels [554].

Current pharmacological trials to treat sarcopenia focus on the potential suitability of various agents, including appetite stimulants, protein anabolic agents, growth hormones, anabolic steroids, androgenic steroids, androgenic receptor modulators, angiotensin-converting enzyme inhibitors, troponin activators, select receptor blockers and myostatin inhibitors [15,560,561,562,563]. Interesting therapeutic options to treat sarcopenia are also provided by interference with atrophy–hypertrophy signaling pathways [564,565,566,567,568]. The above outlined degradation of agrin by neuro-trypsin at the neuromuscular junction [513,514,515] also presents a potential therapeutic target to address abnormal innervation patterns in aged skeletal muscles by employing agrin replacement therapy [516]. One of the most interesting biomedical approaches to treat sarcopenia is myostatin therapy. 

Myostatin is a secreted myogenic factor that acts as a negative regulator of skeletal muscle growth. It belongs to the transforming growth factor TGF-beta family of proteins and functions by inhibiting the phosphorylation of Akt protein kinase within the insulin-like growth factor 1–phosphatidylinositol-3-kinase–serine/threonine protein kinase PKB–mammalian target of rapamycin (IGF-1/PI3K/Akt/mTOR) signaling pathway [569]. Consequently, the inhibition of a negative regulator might result in a positive effect on skeletal muscle growth. This can be supported by (i) natural mechanisms, such as physical exercise, (ii) dietary supplements and nutraceutical agents and/or (iii) pharmacological/biotechnological intervention with myostatin inhibitors [570,571,572], including antibody-based therapy [573]. The rebalancing of muscular atrophy versus hypertrophy by a growth-promoting process could modulate the aging process and have a positive effect on physical fitness and neuromuscular function [574]. 

Ideally, the above-described therapeutic approaches to improve general skeletal muscle strength would especially target the fast-twitching fiber population that is mostly susceptible to muscular atrophy in the elderly [50,339,558]. Figure 6 provides a summary of current therapeutic options to treat sarcopenia of old age. For a critical assessment of current pharmacological strategies to halt or reverse age-related muscular atrophy, see the recent review articles by Cho et al. [15], Kim et al. [561] and Huang et al. [563].

## 5. Conclusions

The proteomic analysis of muscular atrophy in association with sarcopenia has detected distinct changes in a variety of protein families. Alterations in aging skeletal muscles include proteins involved in fiber contraction and relaxation, the regulation of excitation–contraction coupling, ion homeostasis, energy metabolism, maintenance of the cytoskeleton, the extracellular matrix and the cellular stress response. Skeletal muscle aging was shown to be linked to a tendency of fast-to-slow transitions and increased oxidative bioenergetics, as well as myofibrotic changes and a drastic increase in the expression of molecular chaperones. These proteomic findings support the concept of extensive degenerative and adaptive responses in the skeletal musculature due to sarcopenia of old age. Independently verified transcriptomic and proteomic markers of fiber-type shifting and metabolic modifications can now be used as indicators of molecular and cellular changes in both aging human skeletal muscles and animal models of sarcopenia. In the future, it will be of interest to study proteome-wide differences between age-related skeletal muscle wasting and other types of muscular atrophy caused by a variety of diverse triggering factors, such as denervation following motor nerve crush or spinal cord injury, prolonged bedrest in association with chronic disease, inappropriate levels of neuromuscular loading during plaster cast immobilization or prolonged exposure to microgravity. Since skeletal muscle performance deteriorates following extended periods of microgravity [3,575,576], which has been studied by proteomics [577], it has been suggested that certain aspects of neuromuscular alterations during prolonged spaceflights resemble changes in sarcopenia [578]. This opens new possibilities to study accelerated types of muscle-related stress and the molecular and cellular factors involved in muscular atrophy by the exposure of muscle cells to microgravity [579]. The detailed comparison of proteomic and systems bioinformatic data of different forms of muscular atrophy can be helpful to dissect the signaling mechanisms and disturbed biochemical, physiological and cellular processes that lead to diverse forms of muscle wasting.

## Figures and Tables

**Figure 1 ijms-24-02415-f001:**
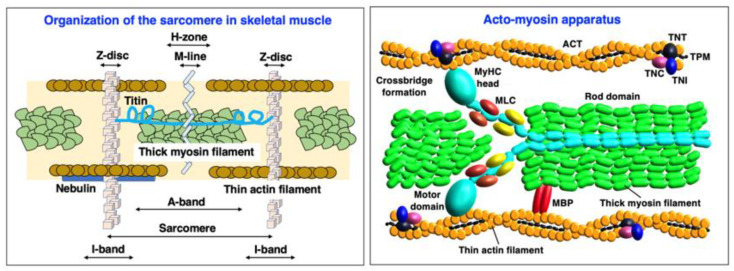
Overview of the sarcomeric structure and the contractile acto-myosin apparatus of skeletal muscle. The diagram to the left outlines the arrangements of the thick myosin-containing filament, the half-sarcomere spanning titin filament and the actin/nebulin-containing thin filament in relation to their positions within the A-band, the I-band, the H-zone, the M-line and the Z-disc structures of the sarcomere. The diagram to the right shows the interactions between the head structure of myosins and filamentous actin molecules that are involved in the crossbridge formation during skeletal muscle contractions. Details of the subunit composition and isoform diversity of myosin heavy chains, myosin light chains, myosin binding proteins, actins, tropomyosins and troponins are given in Figure 2 below. Abbreviations used: ACT, actin; MLC, myosin light chain; MYBP, myosin binding protein; MyHC, myosin heavy chain; TNC; troponin-C; TNI; troponin-I; TNT, troponin-T; TPM, tropomyosin.

**Figure 2 ijms-24-02415-f002:**
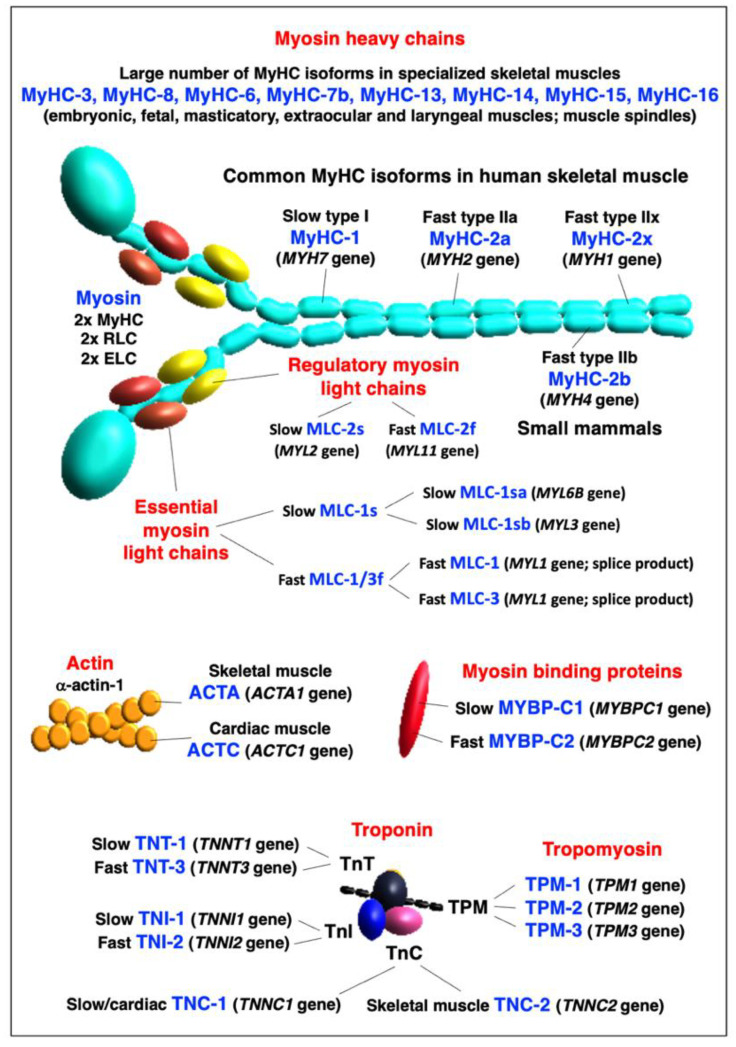
Summary of slow versus fast isoforms of key sarcomeric proteins. The diagrammatic presentation and color coding of individual contractile proteins is identical to the descriptions given in the overview of the sarcomeric structure and the contractile acto-myosin apparatus of skeletal muscle in Figure 1 above. Abbreviations used: ACT, actin; ELC, essential light chain; MYBP, myosin-binding protein; MyHC, myosin heavy chain; MLC, myosin light chain; RLC, regulatory light chain; TNC; troponin-C; TNI; troponin-I; TNT, troponin-T; TPM, tropomyosin.

**Figure 3 ijms-24-02415-f003:**
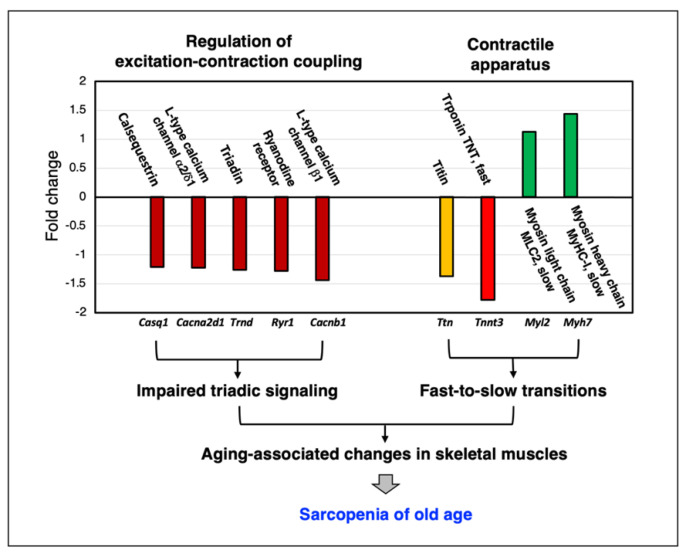
Representative example of the mass-spectrometry-based proteomic analysis of skeletal muscle aging. Shown are crucial regulatory proteins of excitation–contraction coupling and Ca^2+^- homeostasis (dihydropyridine receptor L-type Ca^2+^-channel, ryanodine receptor Ca^2+^-release channel, calsequestrin and triadin) and sarcomeric proteins (titin, troponin, myosin light chain and myosin heavy chain). Mass spectrometric analyses of young versus aged wild-type mouse diaphragm muscle specimens were carried out as previously described in detail [201,269,333].

**Figure 4 ijms-24-02415-f004:**
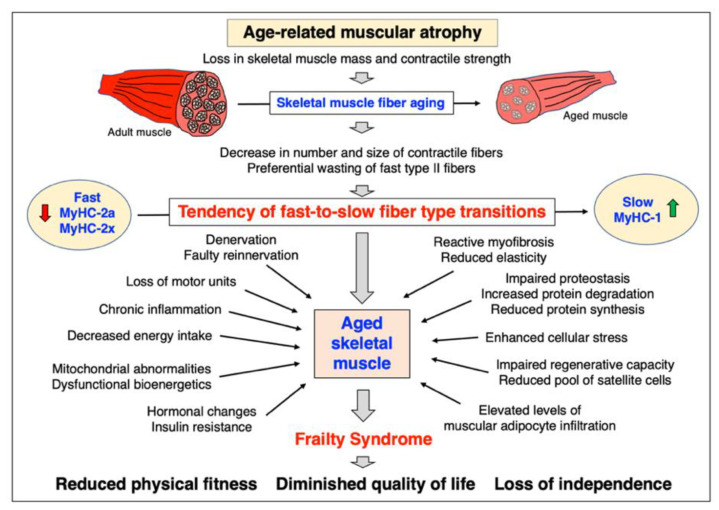
Overview of the multi-factorial changes during human skeletal muscle aging. The higher susceptibility of fast-twitching type II fibers causes a tendency of fast-to-slow transitions in senescent muscles. This is reflected by a switch from fast myosin heavy chain isoforms (MyHC-2a, MyHC-2x) to slower counterparts (MyHC-1) during skeletal muscle aging.

**Figure 5 ijms-24-02415-f005:**
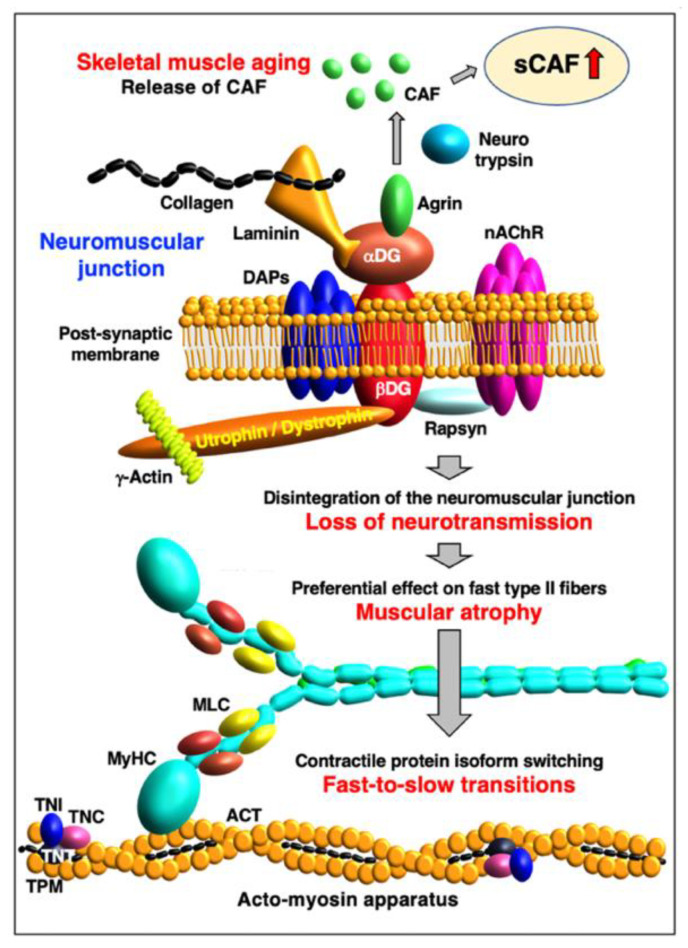
Agrin as a potential serum biomarker of skeletal muscle aging. Shown is the linkage between the disintegration of the neuromuscular junction during skeletal muscle aging and resulting preferential loss of neurotransmission to fast type II fibers. A potential biomarker candidate of this process is the release of carboxy-terminal agrin fragments (CAF) that can be measured in the serum (sCAF) of patients suffering from sarcopenia of old age. At the neuromuscular junction, the proteoglycan agrin associates with the dystroglycan complex (alpha/beta-DG), which forms an integral part of the sub-sarcolemmal utrophin/dystrophin lattice and its associated proteins (DAPs) at the post-synaptic membrane. During skeletal muscle aging, the integrity of the neuromuscular junction is lost, and agrin is proteolytically cleaved by the enzyme neuro-trypsin. This results in the production of distinct agrin fragments that can be conveniently detected in a minimally invasive way in suitable biofluids.

**Figure 6 ijms-24-02415-f006:**
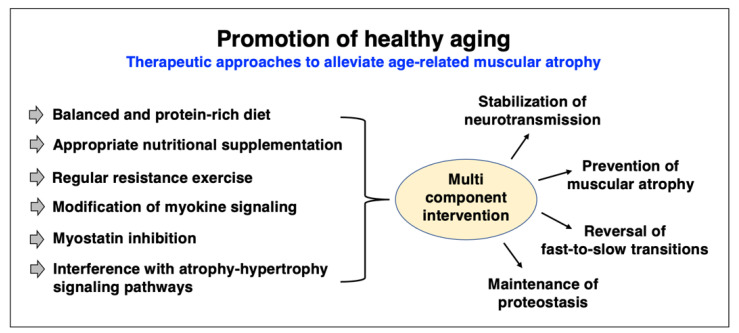
Overview of therapeutic approaches to counteract age-related muscular atrophy.

**Table 1 ijms-24-02415-t001:** Proteomic profiling of key components involved in the contraction–relaxation cycle of mouse diaphragm muscle *.

Contractile Protein	Accession Number/Gene	Coverage/Peptides	Molecular Mass
**Myosin heavy chains (MyHC)**			
MyHC-1, slow muscle (Myosin-7)	Q91Z83/*Myh7*	57.7/139	222.7
MyHC-2x, fast muscle (Myosin-1)	Q5SX40/*MyH1*	69.8/192	223.2
MyHC-2b, fast muscle (Myosin-4)	Q5SX39/*MyH4*	66.0/174	222.7
MyHC-8, perinatal muscle (Myosin-8)	P13542/*MyH8*	46.7/124	222.6
**Myosin light chains (MLC)**			
MLC-1/3, skeletal muscle	P05977/*Myl1*	84.0/19	20.6
MLC-2, skeletal muscle	P97457/*Mylpf*	88.8 /19	18.9
MLC-2, cardiac muscle	P51667/*Myl2*	75.3/12	18.9
MLC-3, skeletal muscle	P09542/*Myl3*	78.4 /16	22.4
**Myosin-binding proteins (MYBP)**			
MYBP-C2, fast-type	Q5XKE0/*Mybpc2*	59.5/51	127.3
MYBP-H	P70402/*Mybph*	25.3/7	52.6
**Actin (ACT) filament**			
Alpha-Actin ACTA, skeletal muscle	P68134/*Acta1*	68.2/25	42.0
F-ACT capping protein, subunit a-2	P47754/*Capza2*	54.9/10	32.9
F-ACT capping protein, subunit b	P47757-2/*Capzab*	33.8/8	30.6
**Tropomyosin (TPM) complex**			
TPM, alpha-1 chain	P58771/*Tpm1*	77.8/37	32.7
TPM, beta chain	P58774/*Tpm2*	76.1/38	32.8
TPM, alpha-3 chain	P21107/*Tpm 3*	68.1/25	33.0
TPM, alpha-4 chain	Q6IRU2/*Tpm 4*	37.9/9	28.5
**Troponin (TN) complex**			
TNI-1, slow skeletal muscle	Q9WUZ5/*Tnni1*	28.9/7	21.7
TNI-2, fast skeletal muscle	P13412/*Tnni2*	44.0/9	21.3
TNT-1, slow skeletal muscle	O88346-3/*Tnnt1*	28.4/8	30.0
TNT-3, fast skeletal muscle	Q9QZ47-12/*Tnnt3*	40.2/13	28.3
TNC-1, slow/cardiac muscle	P19123/*Tnnc1*	47.8/6	18.4
TNC-2, skeletal muscle	P20801/*Tnnc2*	79.4/11	18.1
**Z-disc complex**			
Filamin FLNC	Q8VHX6-2/*Flnc*	42.0/71	287.2
Alpha-Actinin ACTN-2	Q9JI91/*Actn2*	68.3/50	103.8
Alpha-Actinin ACTN-3	O88990/*Actn3*	65.2/48	103.0
Telethonin TCAP	O70548/*Tcap*	36.5/5	19.1
Myozenin MYOZ-1	Q9JK37/*Myoz1*	53.7/7	31.4
Myozenin MYOZ-2	Q9JJW5/*Myoz2*	58.3/12	29.7
Myozenin MYOZ-3	Q8R4E4/*Myoz3*	29.4/5	27.0
**M-line complex**			
Myomesin MYOM-1	Q62234-2/*Myom1*	57.4/70	175.3
Myomesin MYOM-3	A2ABU4/*Myom3*	52.2/47	161.7
Obscurin OBSCN	A2AAJ9/*Obscn*	31.5/135	965.8
**Half-sarcomere-spanning titin filament**			
Titin TNN	A2ASS6/*Ttn*	51.4/1284	3904.1
Muscle ankyrin repeat protein MARP	Q9WV06/*Ankrd2*	26.2/7	36.7

* Mass spectrometric analyses of wild-type mouse diaphragm muscle specimens were carried out as previously described in detail [201,269,333]. The table lists representative protein species that are present in the sarcomere of skeletal muscle fibers.

**Table 2 ijms-24-02415-t002:** List of major mass-spectrometry-based proteomic profiling studies focusing on contractile proteins in aged human skeletal muscle tissue.

Specimens	Bioanalytical Approach	Proteomic Changes	References
*Vastus lateralis* (20–25 years versus 70–76 years)	2D-DIGE, ESI-MS/MS, Pro-Q Diamond, PAGE analysis of MyHC isoforms	Increase in MLC-2s, ACTC and MyHC-I; decrease in MLC2f, TNT-3, TPM-3 and MyHC-2x; shift in phosphorylated MLC-2f to MLC-2s isoforms	Gelfi et al. [401]
*Vastus lateralis* (47–62 years versus 76–82 years)	2D-DIGE, MALDI-TOF, IB	Increase in ACTC; decrease in ACTA, MLC-2, TNT-1 and TNC-1	Staunton et al. [402], Ohlendieck [403]
*Vastus lateralis* (53 years mean age versus 78 years mean age)	Soluble proteins, LC-MS/MS, IB	Increase in MARP/ANKRD2; decreases in MLC-1/3, MyHC-2x and TTN	Théron et al. [202]
*Vastus lateralis* (48–61 years versus 76–82 years post-menopausal women)	2D-GE (CBB), LC-MS/MS, IB	Increase in MARP/ANKRD2, MLC-1/3f, ACTA, TNT-3 and MYOZ-1; decreases in MLC2s and TNN	Gueugneau et al. [404]
*Rectus abdominis* (0–12 years versus 52–76 years)	Oxi-proteome analysis, 2D-GE, protein carbonyl immuno detection	Detection of age-related carbonylation of MyHC-1, MYBP-C1 and TNT-1	Dos Santos et al. [405]
*Vastus lateralis* (18–30 years versus >55 years; trained and untrained)	LC-MS/MS, SRM, PAGE analysis of MyHC isoforms	Increase in MyHC-1; decrease in MyHC-2a; establishment of quantitative differences in myosin light chain composition	Cobley et al. [406]
*Vastus lateralis* (22–27 years versus 65–75 years)	Single-muscle-fiber proteomics, LC-MS/MS	Differential effects on fast versus slow fibers based on MyHC-1, MyHC-2a and MyHC-2x distribution analysis; increase in chaperones of MyHC and ACTA	Murgia et al. [407]
Quadriceps muscle (66–80 years) of healthy versus cancer patients	LC-ESI-MS/MS, SWATH MS, IFM, IB	Differential expression of MyHC-1, MyHC-2a and MyHC-2x in healthy elderly versus cancer patients with or without weight loss	Ebhardt et al. [408]
*Vastus lateralis* (23 years mean age versus 71 years mean age)	2D-GE (CBB), Pro-Q Diamond, MALDI-TOF MS, PAGE analysis of MyHC isoforms, IB	Increase in MyHC-1; decrease in MyHC-2a and MyHC-2x; myosin/actin ratio not affected; differential effects on expression of TNT-3, ACTA and ACTC proteoforms	Brocca et al. [409]
*Vastus lateralis* (Obese and healthy older men of average age 66 undergoing resistance training and energy restriction)	LC-MS/MS, deuterated water labeling of newly synthesized skeletal muscle proteins	Determination of synthesis rate of myofibrillar proteins (MyHC, MLC, ACTA, TPM, TNC, TNT, TNI)	Murphy et al. [221]
*Vastus lateralis* (range of individuals from 20 to 87 years of age)	TMT, LC-MS/MS	Decrease in MYBP-H; switch from MyHC-2x/MyHC-2a to MyHC-1; differential effects on TNT-3, TPM-1 and MYOZ-2 expression	Ubaida-Mohien et al. [410,411]
*Vastus lateralis* (25 years mean age versus 62 years mean age)	LC-MS/MS, PAGE analysis of MyHC isoforms	Reduced acto-myosin abundance; decrease in ACTA and MYBP-H; increase in ACTC and TNT-1	Vann et al. [412]
*Vastus lateralis* (21 years mean age versus 73 years mean age)	2D-GE (CBB), LC-MS/MS, IB	Increase in TNT-1 and MARP; decrease in ACTA, TNT-3 and MYOZ-1	Gueugneau et al. [413]
*Vastus lateralis* (25 years mean age versus 67 years mean age)	iTRAQ, LC-MS/MS	Decrease in ACTA and FLNC	Deane et al. [414]

## Data Availability

Mass spectrometric raw data from studies of aging diaphragm muscle shown in tables and figures are available on request.

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
