# Peer review of "Fiber-Type Shifting in Sarcopenia of Old Age: Proteomic Profiling of the Contractile Apparatus of Skeletal Muscles"

_ijms, 2023, doi:10.3390/ijms24032415_

Round 1

Reviewer 1 Report

RE:             Manuscript: IJMS-2134154

Title: Fiber type shifting in sarcopenia of old age: Proteomic profiling of the contractile apparatus of skeletal muscles

Authors: Dowling et al.

REC.:          Revisions required  

SUMMARY

In this manuscript, Dowling and colleagues reviewed the role of mass spectrometry to identify proteome-wide changes in atrophying skeletal muscles with aging. While an update of the references is needed, the authors have provided a well-written and informative review that indeed will garner some reader interest in the research community.

MAJOR COMMENTS

1.    Please provide a final search and update of the references in sections 2.1.7 and 3.2. For example, in a recent study in “Aging” published in December 2022 (https://doi.org/10.18632/aging.204435), Melov et al. demonstrated the use of RNA-seq and single-nuclei RNA-seq for identifying aging markers (including age-related fiber-type changes) in multiple distinct cell-types from 72 lower limb human muscle biopsies.

2.    While it may be a reader-friendly approach to end the manuscript with therapeutics to combat sarcopenia (discussed in sections 3.1 and 4.3), it is not the focus of the current paper, and one could make an argument to remove it completely.

a.     If you do decide to keep these sections, you must, however, only include evidence from long-term (minimally 3 months - 24 months) double-blinded randomized clinical trials and not prospective trials, unblinded/single-blinded trials, acute interventions or countermeasures that are largely unproven. This comment mainly applies to sections 3.1 and 4.3.  

                                                   i.     Thus, please remove references 352 (Lu et al: study proposal), 528 (dePaula: acute intervention) and 529 (Chow et al: single-blinded study proposal). Collectively, there is no strong evidence to support vibration training to mitigate sarcopenia.

                                                 ii.     In addition, similar arguments can be made for electromyostimulation: While there is indeed more evidence from the Kemmler and Stengel group to combat age-related muscle wasting vs. vibration training, this is still mainly one research group.

                                               iii.     Importantly, there is a matter of access, practicality, and realistic implementation for the aging population, especially during pandemics. Because of COVID-19, elderly across the world were restricted to exercise from home and access to traditional gyms, parks, recreational and rehab facilities were limited. Elderly simply did not, and still don’t, have easy access to advanced strength training equipment and certainly not alternative/expensive rehabilitation modalities.   

In contrast, the combination of low-intensity, home-based resistance exercise (HBRE) and multi-ingredient supplementation appears to be effective in combatting dynapenia and sarcopenia. Some of these published RCTs also include fiber-typing data, which would be more appropriate to include for this review (vs. vibration training & electromyostimulation).

MINOR COMMENTS

1.    Line 99. Fast-twitch?

Author Response

RESPONSE to REVIEWER 1:

Reviewer 1, Comment 1: ‘SUMMARY, In this manuscript, Dowling and colleagues reviewed the role of mass spectrometry to identify proteome-wide changes in atrophying skeletal muscles with aging. While an update of the references is needed, the authors have provided a well-written and informative review that indeed will garner some reader interest in the research community. MAJOR COMMENTS 1. Please provide a final search and update of the references in sections 2.1.7 and 3.2. For example, in a recent study in “Aging” published in December 2022 (https://doi.org/10.18632/aging.204435), Melov et al. demonstrated the use of RNA-seq and single-nuclei RNA-seq for identifying aging markers (including age-related fiber-type changes) in multiple distinct cell-types from 72 lower limb human muscle biopsies’.

Response: We would like to thank Reviewer 1 for the positive evaluation of our manuscript. As suggested, we have carried out a final search of suitable references that cover the proteomic analysis of aged skeletal muscles. The paper by Perez et al. (Perez K, Ciotlos S, McGirr J, Limbad C, Doi R, Nederveen JP, Nilsson MI, Winer DA, Evans W, Tarnopolsky M, Campisi J, Melov S. Single nuclei profiling identifies cell specific markers of skeletal muscle aging, frailty, and senescence. Aging (Albany NY). 2022;14(23):9393-9422) has been added to the revised manuscript, in addition to two recent papers on potential metabolic markers of sarcopenia. The remaining references have been renumbered accordingly. A new paper has been quoted on biofluid analysis of sarcopenia, 3 recently published papers have been added to the part outlining the proteomic analysis of animal models of sarcopenia, and a paper on dynamic proteome profiling (DPP) of cellular aging has been incorporated in the revised manuscript, as outlined below:

New paper quoted in Section 2.1.5. on mass spectrometric analysis. This paper used dynamic proteome profiling (DPP) to study muscle aging in a myoblast cell culture model:

Revised text on Page 8: ‘… The DPP technique was recently applied to study the relative abundance and fractional synthesis rate of proteins in human muscle biopsy specimens [218] and during C2C12 myoblast differentiation [219] and cellular aging [220] …’.

New reference [220]: Brown, A.D.; Stewart, C.E.; Burniston, J.G. Degradation of ribosomal and chaperone proteins is attenuated during the differentiation of replicatively aged C2C12 myoblasts. J. Cachexia Sarcopenia Muscle. 2022, 13, 2562-2575.

New papers quoted on the proteomic analysis of mouse models of sarcopenia:

Revised Page 18: ‘… General alterations in the senescent skeletal muscle proteome, including abundance changes and isoform switching of contractile proteins, were examined in a large number of mass spectrometric surveys using both the well-established rat model of sarcopenia [210,456-473] and aging mouse muscles [474-486] …’.

New references [484] to [486]

[484] Roberts, B.M.; Deemer, S.E.; Smith, D.L. Jr.; Mobley, J.A.; Musi, N.; Plaisance, E.P. Effects of an exogenous ketone ester using multi-omics in skeletal muscle of aging C57BL/6J male mice. Front. Nutr. 2022, 9, 1041026.

[485] Campbell, M.D.; Martín-Pérez, M.; Egertson, J.D.; Gaffrey, M.J.; Wang, L.; Bammler, T.; Rabinovitch, P.S.; MacCoss, M.; Qian, W.J.; Villen, J.; Marcinek, D. Elamipretide effects on the skeletal muscle phosphoproteome in aged female mice. Geroscience. 2022, 44, 2913-2924.

[486] Jessica Lo, H.T.; Yiu, T.L.; Wang, Y.; Feng, L.; Li, G.; Lui, M.P.; Lee, W.Y. Fetal muscle extract improves muscle function and performance in aged mice. Front. Physiol. 2022, 13, 816774.

New paper quoted on biofluid analysis of sarcopenia:

Revised text on Page 21: ‘… The serum of both older humans suffering from sarcopenia [504-507] and senescent mice [508] were shown to exhibit differential changes in common markers that are associated with inflammation, remodeling of the extracellular matrix and mitochondrial functions [398] …’.

New reference [507] Wu, J.; Cao, L,. Wang, J.; Wang, Y.; Hao, H.; Huang, L. Characterization of serum protein expression profiles in the early sarcopenia older adults with low grip strength: a cross-sectional study. BMC Musculoskelet. Disord. 2022, 23, 894.

Additional papers on gene expression and metabolomic studies in aged human muscles:

Revised Page 21: ‘... Ideally, proteomic findings are correlated to the results from systematic transcriptomic and metabolomic studies of sarcopenia [510-512] …’.

New references [510] to [512]:

[510] Perez, K.; Ciotlos, S.; McGirr, J.; Limbad, C.; Doi, R.; Nederveen, J.P.; Nilsson, M.I.; Winer, D.A.; Evans, W.; Tarnopolsky, M.; et al. Single nuclei profiling identifies cell specific markers of skeletal muscle aging, frailty, and senescence. Aging (Albany NY). 2022, 14, 9393-9422.

[511] Zhao, Q.; Shen, H.; Liu, J.; Chiu, C.Y.; Su, K.J.; Tian, Q.; Kakhniashvili, D.; Qiu, C.; Zhao, L.J.; Luo, Z.; Deng, H.W. Pathway-based metabolomics study of sarcopenia-related traits in two US cohorts. Aging (Albany NY). 2022, 14, 2101-2112.

[512] Tsai, J.S.; Wang, S.Y.; Chang, C.H.; Chen, C.Y.; Wen, C.J.; Chen, G.Y.; Kuo, C.H.; Tseng, Y.J.; Chen, C.Y. Identification of traumatic acid as a potential plasma biomarker for sarcopenia using a metabolomics-based approach. J. Cachexia Sarcopenia Muscle. 2022, 13, 276-286.

Reviewer 1, Comment 2: ‘2. While it may be a reader-friendly approach to end the manuscript with therapeutics to combat sarcopenia (discussed in sections 3.1 and 4.3), it is not the focus of the current paper, and one could make an argument to remove it completely. (a). If you do decide to keep these sections, you must, however, only include evidence from long-term (minimally 3 months - 24 months) double-blinded randomized clinical trials and not prospective trials, unblinded/single-blinded trials, acute interventions or countermeasures that are largely unproven. This comment mainly applies to sections 3.1 and 4.3.  (i). Thus, please remove references 352 (Lu et al: study proposal), 528 (dePaula: acute intervention) and 529 (Chow et al: single-blinded study proposal). Collectively, there is no strong evidence to support vibration training to mitigate sarcopenia’.

Response: We agree and have removed the original references [352], [528] and [529]. On Page 15, original reference [352] was substituted with original reference [527] on resistance training. However, the reference numbering has been changed due to the introduction of a new reference [220] as outlined above. We have also removed, as outlined in response to below point, the text and references relating to electromyostimulation therapy. Thus, the revised manuscript does no longer discuss vibration and electrostimulation therapy, and these points were also removed from revised Figure 6. Original references [352] and [528] to [533] were removed from revised Pages 22-23. The remaining references have been renumbered accordingly. The list of suitable references on the therapy of muscular atrophy and muscle aging were updated and now include 3 new references, as outlined below:

Revised Page 15 (lines 665-669): ‘... Thus, to counteract the age-dependent decline in skeletal muscle performance, optimized rehabilitation [350] and appropriate physical exercise regimes, such as moderate resistance exercises [351-353], are crucial to minimize oxidative stress and inflammation in sarcopenia [354,355] …’.

Moved reference [353]: de Sá Souza, H.; de Melo, C.M.; Piovezan, R.D.; Miranda, R.E.E.P.C.; Carneiro-Junior, M.A.; Silva, B.M.; Thomatieli-Santos, R.V.; Tufik, S.; Poyares, D.; D'Almeida, V. Resistance Training Improves Sleep and Anti-Inflammatory Parameters in Sar-copenic Older Adults: A Randomized Controlled Trial. Int. J. Environ. Res. Public Health. 2022, 19, 16322.

Revised Page 26: ‘… Current pharmacological trials to treat sarcopenia focus on the potential suitability of various agents, including appetite stimulants, protein anabolic agents, growth hor-mones, anabolic steroids, androgenic steroids, androgenic receptor modulators, angio-tensin-converting enzyme inhibitors, troponin activators, select receptor blockers and myostatin inhibitors [15, 552-555]…’.

New references [552], [554] and [555]:

[552] Li, Y.; Chen, M.; Zhao, Y.; Li, M.; Qin, Y.; Cheng, S.; Yang, Y.; Yin, P.; Zhang, L.; Tang, P. Advance in Drug Delivery for Ageing Skeletal Muscle. Front. Pharmacol. 2020, 11, 1016.

[554] Canfora, I.; Tarantino, N.; Pierno, S. Metabolic Pathways and Ion Channels Involved in Skeletal Muscle Atrophy: A Starting Point for Potential Therapeutic Strategies. Cells. 2022, 11, 2566.

[555] Huang, L.; Li, M.; Deng, C.; Qiu, J.; Wang, K.; Chang, M.; Zhou, S.; Gu, Y.; Shen, Y.; Wang, W.; et al. Potential Therapeutic Strategies for Skeletal Muscle Atrophy. Antioxidants. 2023, 12, 44.

Reviewer 1, Comment 3: ‘(ii). In addition, similar arguments can be made for electromyostimulation: While there is indeed more evidence from the Kemmler and Stengel group to combat age-related muscle wasting vs. vibration training, this is still mainly one research group’.

Response: We agree and have removed the discussion of electrostimulation as a potential therapy of sarcopenia, which also included the removal of original references [530] to [533] and a revision of Figure 6.

Reviewer 1, Comment 4: ‘(iii). Importantly, there is a matter of access, practicality, and realistic implementation for the aging population, especially during pandemics. Because of COVID-19, elderly across the world were restricted to exercise from home and access to traditional gyms, parks, recreational and rehab facilities were limited. Elderly simply did not, and still don’t, have easy access to advanced strength training equipment and certainly not alternative/expensive rehabilitation modalities’.

Response: We would like to thank Reviewer 1 for pointing out these issues and have added this information to revised Section 4.3:

Revised Page 22: ‘… to avoid the premature loss of physical strength and muscle mass. However, a crucial issue for the elderly is proper access to advanced strength training equipment and the realistic implementation of health promoting support structures, especially during pandemics. During the current COVID-19 crisis, the aged population has only limited access to gyms, recreational facilities and rehabilitation services, and this situation has to be urgently addressed to promote healthy aging …’.

Reviewer 1, Comment 5: ‘In contrast, the combination of low-intensity, home-based resistance exercise (HBRE) and multi-ingredient supplementation appears to be effective in combatting dynapenia and sarcopenia. Some of these published RCTs also include fiber-typing data, which would be more appropriate to include for this review (vs. vibration training & electromyostimulation)’.

Response: We agree and have added a new paragraph with relevant references on exercise-related fibre type shifting.

Revised Page 23: ‘… At advanced age, combining a low-intensity form of home-based resistance exercise with proper nutrition and a multi-ingredient supplementation seems to be the most effective way to treat sarcopenia. …………. Regular exercise has a profound effect on the skeletal muscle proteome [68,354,419] and muscle fiber type diversification [541-543]. Especially resistance exercise aimed at improving the contractile strength of aged skeletal muscles is generally associated with alterations in myofiber size, muscle re-innervation, fiber type-specific myonuclear adaptations, mitochondrial remodelling and fibre type shifting [544-549]. Since the age-related loss of skeletal muscle mass is mostly due to a drastic reduction in the size of fast-twitching type 2 fibers [50,51,339,550], it is encouraging that resistance exercise specifically results in the hypertrophy of type 2 muscle fibres, although it does not appear to affect patterns of fiber type grouping in aged muscles [551]. Distinct changes in MyHC isoform expression patterns are usually exemplified by reduced MyHC-1 and increased MyHC-2x levels [546] …’.

New references [541] to [551]:

[541] Yan, Z.; Okutsu, M.; Akhtar, Y.N.; Lira, V.A. Regulation of exercise-induced fiber type transformation, mitochondrial biogenesis, and angiogenesis in skeletal muscle. J. Appl. Physiol. 2011, 110, 264-274.

[542] Wilson, J.M.; Loenneke, J.P.; Jo, E.; Wilson, G.J.; Zourdos, M.C.; Kim, J.S. The effects of endurance, strength, and power training on muscle fiber type shifting. J. Strength Cond. Res. 2012, 26, 1724-1729.

[543] Qaisar, R.; Bhaskaran, S.; Van Remmen, H. Muscle fiber type diversification during exercise and regeneration. Free Radic. Biol. Med. 2016, 98, 56-67.

[544] Coletti, C.; Acosta, G.F.; Keslacy, S.; Coletti, D. Exercise-mediated reinnervation of skeletal muscle in elderly people: An up-date. Eur. J. Transl. Myol. 2022, 32, 10416.

[545] Blocquiaux, S.; Gorski, T.; Van Roie, E.; Ramaekers, M.; Van Thienen, R.; Nielens, H.; Delecluse, C.; De Bock, K.; Thomis, M. The effect of resistance training, detraining and retraining on muscle strength and power, myofibre size, satellite cells and myonuclei in older men. Exp. Gerontol. 2020, 133, 110860.

[546] Miller, M.S.; Callahan, D.M.; Tourville, T.W.; Slauterbeck, J.R.; Kaplan, A.; Fiske, B.R.; Savage, P.D.; Ades, P.A.; Beynnon, B.D.; Toth, M.J. Moderate-intensity resistance exercise alters skeletal muscle molecular and cellular structure and function in inactive older adults with knee osteoarthritis. J. Appl. Physiol. 2017, 122, 775-787.

[547] Mesquita, P.H.C.; Lamb, D.A.; Parry, H.A.; Moore, J.H.; Smith, M.A.; Vann, C.G.; Osburn, S.C.; Fox, C.D.; Ruple, B.A.; Huggins, K.W.; et al.. Acute and chronic effects of resistance training on skeletal muscle markers of mitochondrial remodeling in older adults. Physiol. Rep. 2020, 8, e14526.

[548] Moro, T.; Brightwell, C.R.; Volpi, E.; Rasmussen, B.B.; Fry, C.S. Resistance exercise training promotes fiber type-specific myo-nuclear adaptations in older adults. J. Appl. Physiol. 2020, 128, 795-804.

[549] Fry, C.S.; Noehren, B.; Mula, J.; Ubele, M.F.; Westgate, P.M.; Kern, P.A.; Peterson, C.A. Fibre type-specific satellite cell response to aerobic training in sedentary adults. J. Physiol. 2014 , 592, 2625-2635.

[550] Nilwik, R.; Snijders, T.; Leenders, M.; Groen, B.B.; van Kranenburg, J.; Verdijk, L.B.; van Loon, L.J. The decline in skeletal muscle mass with aging is mainly attributed to a reduction in type II muscle fiber size. Exp. Gerontol. 2013, 48, 492-498.

[551] Kraková, D.; Holwerda, A.M.; Betz, M.W.; Lavin, K.M.; Bamman, M.M.; van Loon, L.J.C.; Verdijk, L.B.; Snijders, T. Muscle fiber type grouping does not change in response to prolonged resistance exercise training in healthy older men. Exp. Gerontol. 2023 (in press) doi: 10.1016/j.exger.2023.112083.

Revised text on Page 23: ‘… Interesting therapeutic options to treat sarcopenia are also provided by the interference with atrophy-hypertrophy signaling pathways [556-560] …’.

New references [556] to [560]:

[556] Sartori, R.; Romanello, V.; Sandri, M. Mechanisms of muscle atrophy and hypertrophy: implications in health and disease. Nat. Commun. 2021, 12, 330.

[557] Guo, M.; Yao, J.; Li, J.; Zhang, J.; Wang, D.; Zuo, H.; Zhang, Y.; Xu, B.; Zhong, Y.; Shen, F.; et al. Irisin ameliorates age-associated sarcopenia and metabolic dysfunction. J. Cachexia Sarcopenia Muscle. 2022 (in press) doi: 10.1002/jcsm.13141.

[558] Vainshtein, A.; Sandri, M. Signaling Pathways That Control Muscle Mass. Int. J. Mol. Sci. 2020, 21, 4759.

[559] Penniman, C.M.; Bhardwaj, G.; Nowers, C.J.; Brown, C.U.; Junck, T.L.; Boyer, C.K.; Jena, J.; Fuqua, J.D.; Lira, V.A.; O'Neill, B.T. Loss of FoxOs in muscle increases strength and mitochondrial function during aging. J. Cachexia Sarcopenia Muscle. 2022 (in press) doi: 10.1002/jcsm.13124.

[560] Sirago, G.; Picca, A.; Calvani, R.; Coelho-Júnior, H.J.; Marzetti, E. Mammalian Target of Rapamycin (mTOR) Signaling at the Crossroad of Muscle Fiber Fate in Sarcopenia. Int. J. Mol. Sci. 2022, 23, 13823.

Changes in Figure 6:  The terms ‘vibration therapy’ and ‘electromyostimulation’ were removed, as suggested by Reviewer 1; and instead the terms ‘Modification of myokine signaling’ and ‘Interference with atrophy-hypertrophy signaling pathways’ added to the revised figure.

Reviewer 1, Comment 6: ‘MINOR COMMENTS, 1. Line 99. Fast-twitch?

Response: To address this point, we have added this information on ‘fast-to-slow fiber type shifting’ to the last sentence of the Introduction section as follows:

Revised Page 2: “… A final section of this review briefly summarizes the main factors that are involved in age-related muscular atrophy and associated fast-to-slow fiber type shifting, as well as describes recent progress in biomarker discovery for monitoring muscle aging and the development of novel therapeutic approaches to treat sarcopenia of old age’.

Reviewer 2 Report

This is a very well written review on a topic that is highly pertinent to health in the elderly. It articulates this relevance very well.  There is a comprehensive and useful explanation of the technical aspects of proteomics, including the most up-to-date developments (single cell proteomics and aptamer-based technologies).  Muscle physiology, as germane to the article’s focus, is covered clearly and comprehensively.  Either of these two sections on their own would be very useful, I think, to researchers wanting to gain a thorough level of knowledge about one or other topic.  The review then follows with a clear, relevant and up to date exposition of how proteomics data has informed and is informing the field.

In more detail: Examination of all aspects of the proteomics pipeline seems comprehensive and is also very accessible. I know that many researchers have scant knowledge of the principles and practicalities of proteomics and think that this review would be a very useful to any researcher (whether the context be for the study of muscle or not) to gain a good level of knowledge about the technique.  I am nor an expert in proteomic techniques but was initially surprised to see a section on the importance of 2D gel electrophoresis, which I would have assumed to now be made redundant by new advancements.  However, I found in the literature opinion, and convincing argument, that the technique still has currency (https://www.ncbi.nlm.nih.gov/pmc/articles/PMC7563651/

 and then saw that the authors cite the same article.  However, it is cited in the review in general reference to the technique, rather than to validate ongoing importance. I wonder if it may be worth explicitly addressing the point that 2D gel electrophoresis, though quite an old technique, has still not been superseded for some applications.

The section on muscle physiology/fibre type specification is excellent.  This is followed by a clear section on how features of muscle structure and function change with ageing and contribute to frailty and other aspects of poor health.  The advancements made using proteomics are then clearly and - it appears -comprehensively explained, including reference to animal models.  Finally, there is a forward-looking perspective on biomarker discovery and future therapeutic potential.

I think this is an excellent and very full review in its current format. Other than possibly explaining to those readers who, like me, may be surprised that 2D gel electrophoresis is still very relevant to the technology, the only addition that may be worth considering is reverence to the interesting human microgravity studies, which to date have been more focused on the health of astronauts (e.g. https://www.ncbi.nlm.nih.gov/pmc/articles/PMC6707384/  but being considered as a model for human ageing too (albeit, to my knowledge, currently using only cell culture models; e.g.  https://pubmed.ncbi.nlm.nih.gov/34562651/).  I see that the very last paragraph mentions microgravity/bedrest etc, though, which perhaps this is an adequate hook for readers wanting to find out more.

Author Response

RESPONSE to REVIEWER 2:

Reviewer 2, Comment 1: ‘Comments and Suggestions for Authors: This is a very well written review on a topic that is highly pertinent to health in the elderly. It articulates this relevance very well.  There is a comprehensive and useful explanation of the technical aspects of proteomics, including the most up-to-date developments (single cell proteomics and aptamer-based technologies).  Muscle physiology, as germane to the article’s focus, is covered clearly and comprehensively.  Either of these two sections on their own would be very useful, I think, to researchers wanting to gain a thorough level of knowledge about one or other topic.  The review then follows with a clear, relevant and up to date exposition of how proteomics data has informed and is informing the field. … In more detail: Examination of all aspects of the proteomics pipeline seems comprehensive and is also very accessible. I know that many researchers have scant knowledge of the principles and practicalities of proteomics and think that this review would be a very useful to any researcher (whether the context be for the study of muscle or not) to gain a good level of knowledge about the technique.  I am nor an expert in proteomic techniques but was initially surprised to see a section on the importance of 2D gel electrophoresis, which I would have assumed to now be made redundant by new advancements.  However, I found in the literature opinion, and convincing argument, that the technique still has currency (https://www.ncbi.nlm.nih.gov/pmc/articles/PMC7563651/) and then saw that the authors cite the same article.  However, it is cited in the review in general reference to the technique, rather than to validate ongoing importance. I wonder if it may be worth explicitly addressing the point that 2D gel electrophoresis, though quite an old technique, has still not been superseded for some applications’.

Response: We would like to thank Reviewer 2 for the positive evaluation of our manuscript. As suggested, we have revised the text in Section 2.1.1 on 2D-GE to better emphasise the paper by Marcus et al. (Marcus, K.; Lelong, C.; Rabilloud, T. What Room for Two-Dimensional Gel-Based Proteomics in a Shotgun Proteomics World? Proteomes. 2020, 8, 17) that argues in favour of 2D-GE being still a useful protein separation method in the proteomics analysis pipeline. Especially the initial steps of top-down proteomics with a focus on the analysis of isolated and intact proteoforms can be conveniently performed with 2D-GE techniques.

Revised Page 3 (lines 125-132): ‘… profiling of proteoforms [61]. Although current proteomic analyses use mostly gel-free systems for the initial protein separation step, 2D-GE has not been superseded by chromatographical techniques for specialized applications in top-down proteomics [82]. 2D-GE is still a highly useful protein separation method that plays a key role in many proteomics analysis pipelines that focus on the identification and characterization of isolated and intact proteoforms [52,61]. The 2D-GE based separation step is especially beneficial in the field of applied myology for analyzing the highly diverse array of isoforms of contractile proteins [65-67].  The large-scale survey of …’.

Reviewer 2, Comment 2: ‘The section on muscle physiology/fibre type specification is excellent.  This is followed by a clear section on how features of muscle structure and function change with ageing and contribute to frailty and other aspects of poor health.  The advancements made using proteomics are then clearly and - it appears - comprehensively explained, including reference to animal models.  Finally, there is a forward-looking perspective on biomarker discovery and future therapeutic potential. … I think this is an excellent and very full review in its current format. Other than possibly explaining to those readers who, like me, may be surprised that 2D gel electrophoresis is still very relevant to the technology, the only addition that may be worth considering is reverence to the interesting human microgravity studies, which to date have been more focused on the health of astronauts (e.g. https://www.ncbi.nlm.nih.gov/pmc/articles/PMC6707384/ but being considered as a model for human ageing too (albeit, to my knowledge, currently using only cell culture models; e.g.  https://pubmed.ncbi.nlm.nih.gov/34562651/).  I see that the very last paragraph mentions microgravity/bedrest etc, though, which perhaps this is an adequate hook for readers wanting to find out more’.

Response: We would like to thank Reviewer 2 for the positive evaluation of the section on muscle physiology and fibre type specification. We agree with the point on microgravity studies and aging models, and have accordingly added information and references on this emerging field in muscular atrophy and gerontology research. This includes the 2 references suggested by Reviewer 2 and papers focusing on the analysis of the effects of prolonged bed rest and spaceflight.

Revised Page 24 (Conclusions section): “… exposure to microgravity. Since skeletal muscle performance deteriorates following extended periods of microgravity [3, 567, 568], which has been studied by proteomics [569], it has been suggested that certain aspects of neuromuscular alterations during prolonged spaceflights resemble changes in sarcopenia [570]. This opens new possibilities to study accelerated types of muscle-related stress and the molecular and cellular factors involved in muscular atrophy by exposure of muscle cells to microgravity [571]. The detailed comparison of proteomic …’. 

New references [567] to [571]:

[567] Winnard, A.; Scott, J.; Waters, N.; Vance, M.; Caplan, N. Effect of Time on Human Muscle Outcomes During Simulated Microgravity Exposure Without Countermeasures-Systematic Review. Front. Physiol. 2019, 10, 1046.

[568] Lee, P.H.U.; Chung, M.; Ren, Z.; Mair, D.B.; Kim, D.H. Factors mediating spaceflight-induced skeletal muscle atrophy. Am. J. Physiol. Cell Physiol. 2022, 322, C567-C580.

[569] Schulz, H.; Strauch, S.M.; Richter, P.; Wehland, M.; Krüger, M.; Sahana, J.; Corydon, T.J.; Wise, P.; Baran, R.; Lebert, M.; Grimm, D. Latest knowledge about changes in the proteome in microgravity. Expert Rev. Proteomics. 2022, 19, 43-59.

[570] Cannavo, A.; Carandina, A.; Corbi, G.; Tobaldini, E.; Montano, N.; Arosio, B. Are Skeletal Muscle Changes during Prolonged Space Flights Similar to Those Experienced by Frail and Sarcopenic Older Adults? Life (Basel). 2022, 12, 2139.

[571] Takahashi, H.; Nakamura, A.; Shimizu, T. Simulated microgravity accelerates aging of human skeletal muscle myoblasts at the single cell level. Biochem. Biophys. Res. Commun. 2021, 578, 115-121.

Reviewer 3 Report

Manuscript ID: ijms-2134154

Title: Fiber type shifting in sarcopenia of old age: Proteomic profiling of the contractile apparatus of skeletal muscles

Journal: International Journal of Molecular Sciences

The authors have investigated the Fiber type shifting in sarcopenia of old age: Proteomic profiling of the contractile apparatus of skeletal muscles. Studies with senescent animal models, including mostly aged rodent skeletal muscles, have confirmed fiber-type shifting. The proteomic analysis of fast versus slow isoforms of key contractile proteins, such as myosin heavy chains, myosin light chains, actins, troponins, and tropomyosins, suggests them as suitable bioanalytical tools of fiber-type transitions during aging. The authors have provided a promising prospect regarding myosin heavy chains, myosin light chains, actins, troponins, and tropomyosins as suitable bioanalytical tools for fiber-type transitions. However, some details need further explanation. The article is well-written and well-organized, and I believe that it is suitable for publication in the International Journal of Molecular Sciences. However, I recommend this manuscript can be published after a minor revision and improvement.

1. The author should follow our journal "International Journal of Molecular Sciences" formatting style.

2. The table should be with an advanced style, such as 3 horizontal lines, like one 2 at the top and 1 at the bottom.

3. Please correct typos, errors, and grammar throughout the manuscript; also, the manuscript needs consistency regarding writing and formatting. 

Author Response

RESPONSE to REVIEWER 3:

Reviewer 3, Comment 1: ‘The authors have investigated the Fiber type shifting in sarcopenia of old age: Proteomic profiling of the contractile apparatus of skeletal muscles. Studies with senescent animal models, including mostly aged rodent skeletal muscles, have confirmed fiber-type shifting. The proteomic analysis of fast versus slow isoforms of key contractile proteins, such as myosin heavy chains, myosin light chains, actins, troponins, and tropomyosins, suggests them as suitable bioanalytical tools of fiber-type transitions during aging. The authors have provided a promising prospect regarding myosin heavy chains, myosin light chains, actins, troponins, and tropomyosins as suitable bioanalytical tools for fiber-type transitions. However, some details need further explanation. The article is well-written and well-organized, and I believe that it is suitable for publication in the International Journal of Molecular Sciences. However, I recommend this manuscript can be published after a minor revision and improvement.

Response: We would like to thank Reviewer 3 for the positive evaluation of our manuscript.

Reviewer 3, Comment 2: ‘1. The author should follow our journal "International Journal of Molecular Sciences" formatting style’.

Response: Despite using the IJMS template, unfortunately a certain degree of reformatting appears to occur during the submission process with Macintosh-generated files. This especially affects the table format. To address this issue, we have used the latest version of the IJMS template supplied by the editorial office for our re-submission and have moved the file for revisions from an older MacBook to a newer iMac computer with a macOS Monterey version 12.1 operating system. This will hopefully prevent major re-formatting issues during re-submission.

Reviewer 3, Comment 3: ‘2. The table should be with an advanced style, such as 3 horizontal lines, like one 2 at the top and 1 at the bottom’.

Response: We have used an iMac computer for the revisions and the R1-file displays the proper table formatting as supplied with the IJMS template. However, certain reformatting issues may occur during Macintosh-to-PC conversions and/or the uploading of Macintosh-generated files.

Reviewer 3, Comment 4: ‘3. Please correct typos, errors, and grammar throughout the manuscript; also, the manuscript needs consistency regarding writing and formatting.

Response: In response to this point, an external person from the university language centre has been asked to proofread the revised manuscript text for consistency and errors.

Round 2

Reviewer 1 Report

I would like to commend the authors for removing the references and therapeutic regimens that were largely unproven in the last draft. While I think it is best to references the added sentences to COVID-19 limits on access & home-based training and MIS in the last paragraph, the points are made. 

Author Response

RESPONSE to REVIEWER 1 to re-revision of R1-version:

Reviewer 1, Comment: ‘I would like to commend the authors for removing the references and therapeutic regimens that were largely unproven in the last draft. While I think it is best to references the added sentences to COVID-19 limits on access & home-based training and MIS in the last paragraph, the points are made’.

Response: We would like to thank Reviewer 1 for the positive evaluation of our revised manuscript. We agree with the point on adding references in the context of COVID-19 and sarcopenia, as outlined below. The R2 version has been revised on Page 23 with the addition of suitable references and an extra sentence. The re-revised R2 manuscript file with track changes, submitted as a supplementary file, highlights changed parts in the text and reference section in response to Round 2 comments in BLUE, and the changes made during Round 1 are highlighted in YELLOW. References have been re-numbered accordingly.

Revised Page 23: ‘… to avoid the premature loss of physical strength and muscle mass. However, a crucial issue for the elderly is proper access to advanced strength training equipment and the realistic implementation of health promoting support structures, especially during pandemics. During the current COVID-19 crisis, the aged population has only limited access to gyms, parks, recreational facilities and rehabilitation services causing long-term negative effects on muscle health [533-535], and this situation has to be urgently addressed to promote healthy aging. In addition, the treatment of acute sarcopenia in patients with or without COVID-19 infection has been complicated by the restricted access to health services during the pandemic [536-539] and the increased application of mechanical ventilation and complications during ventilator weaning has caused considerable side effects including skeletal muscle wasting [540]’. 

New references [533] to [540]:

[533] Kirwan, R.; McCullough, D.; Butler, T.; Perez de Heredia, F.; Davies, I.G.; Stewart, C. Sarcopenia during COVID-19 lockdown restrictions: long-term health effects of short-term muscle loss. Geroscience. 2020, 42, 1547-1578.

[534] Shur, N.F.; Creedon, L.; Skirrow, S.; Atherton, P.J.; MacDonald, I.A.; Lund, J.; Greenhaff, P.L. Age-related changes in muscle architecture and metabolism in humans: The likely contribution of physical inactivity to age-related functional decline. Age-ing Res. Rev. 2021, 68, 101344.

[535] Demonceau, C.; Beaudart, C.; Reginster, J.Y.; Veronese, N.; Bruyère, O. The interconnection between Covid-19, sarcopenia and lifestyle. Maturitas. 2022, S0378-5122(22)00204-3.

[536] Welch, C.; Greig, C.; Masud, T.; Wilson, D.; Jackson, T.A. COVID-19 and Acute Sarcopenia. Aging Dis. 2020, 11, 1345-1351. 

[537] Soares, M.N.; Eggelbusch, M.; Naddaf, E.; Gerrits, K.H.L; van der Schaaf, M.; van den Borst, B.; Wiersinga, W.J.; van Vugt, M.; Weijs, P.J.M.; Murray, A.J.; Wüst, R.C.I. Skeletal muscle alterations in patients with acute Covid-19 and post-acute sequelae of Covid-19. J. Cachexia Sarcopenia Muscle. 2022, 13, 11-22.

[538] Piotrowicz, K.; GÄ…sowski, J.; Michel, J.P.; Veronese, N. Post-COVID-19 acute sarcopenia: physiopathology and management. Aging Clin. Exp. Res. 2021, 33, 2887-2898. 

[539] Wierdsma, N.J.; Kruizenga, H.M.; Konings, L.A.; Krebbers, D.; Jorissen, J.R.; Joosten, M.I.; van Aken, L.H.; Tan, F.M.; van Bodegraven, A.A.; Soeters, M.R.; Weijs, P.J. Poor nutritional status, risk of sarcopenia and nutrition related complaints are prev-alent in COVID-19 patients during and after hospital admission. Clin. Nutr. ESPEN. 2021, 43, 369-376. 

[540] Damanti, S.; Cristel, G.; Ramirez, G.A.; Bozzolo, E.P.; Da Prat, V.; Gobbi, A.; Centurioni, C.; Di Gaeta, E.; Del Prete, A.; Calabrò, M.G.; et al. Influence of reduced muscle mass and quality on ventilator weaning and complications during intensive care unit stay in COVID-19 patients. Clin. Nutr. 2022, 41, 2965-2972.